# A global analysis of the fractal properties of clouds revealing anisotropy of turbulence across scales

Karlie N. Rees[1], Timothy J. Garrett[1], Thomas D. DeWitt[1], Corey Bois[1], Steven K. Krueger[1], and Jérôme C. Riedi[2]

[1]Department of Atmospheric Sciences, University of Utah, 135 S 1460 E Rm 819, Salt Lake City, UT 84112, USA
[2]University of Lille, CNRS, UMR 8518 – LOA – Laboratoire d'Optique Atmosphérique, 59000 Lille, France

**Correspondence:** Tim Garrett (tim.garrett@utah.edu)

**Abstract.** The deterministic motions of clouds and turbulence, despite their chaotic nature, nonetheless have been shown to follow simple statistical power-law scalings: a fractal dimension $D$ relates individual cloud perimeters $p$ to measurement resolution, and turbulent fluctuations scale with air parcel separation distance through the Hurst exponent $\mathcal{H}$. However, it remains uncertain whether atmospheric turbulence is best characterized by a split isotropy that is three-dimensional with $\mathcal{H} = 1/3$ at small scales and two-dimensional with $\mathcal{H} = 1$ at large scales, or by a wide-range anisotropic scaling with an intermediate value of $\mathcal{H}$. Here, we introduce an "ensemble fractal dimension" $D_e$ – analogous to $D$ – that relates the total cloud perimeter per domain area $\mathcal{P}$ as seen from space to the measurement resolution, and we show theoretically how turbulent dimensionality and cloud edge geometry can be linked through $\mathcal{H} = D_e - 1$. Observationally and numerically, we find the scaling $D_e \sim 5/3$, or $\mathcal{H} \sim 2/3$, spanning 5 orders of magnitude of scale. Remarkably, the same scaling relationship links two "limiting case" estimates of $\mathcal{P}$ evaluated at resolutions corresponding to the planetary scale and the Kolmogorov microscale suggesting extension to 10 orders of magnitude. Our results are nearly consistent with a previously proposed "23/9D" anisotropic turbulent scaling and suggest that the geometric characteristics of clouds and turbulence in the atmosphere can be easily tied to well-known planetary physical parameters.

## 1 Introduction

The Earth system is radiatively open and materially closed. Radiatively, Earth's global mean temperature is sustained by a balance between absorption of high-intensity shortwave sunlight and the reemission at longwave frequencies to the cold of space. Materially, the total dry atmospheric mass is confined to the planet by gravity and can only be redistributed by turbulent circulations that mix air over a broad range of scales within the thin atmospheric layer. Clouds play important roles in determining the magnitude of both categories of flow. Geometrically speaking, cloud areas govern radiative fluxes (Zelinka et al., 2022) while the edge length or perimeter of clouds controls material exchanges of air between clouds and their clear-sky environment (Zhao and Austin, 2005; Heus et al., 2008; Garrett et al., 2018).

A scientific challenge is that the seemingly objective properties of cloud area and perimeter are a function of the more subjective choice of spatial resolution $\xi$ (defined as either the pixel side length in a satellite image or the grid spacing in a

model). Clouds smaller than $\xi$ cannot be resolved, and the square shapes in a resolved grid do not reflect more irregular cloud structures. Even casual observations of the sky show cloud edges that are intricately complex for any plausibly resolvable scale. For example, the boundary of a small cumulus cloud may appear smoothly rounded at first glance, but fine turbulent structures become apparent when it is viewed through binoculars. The change of observational scale lengthens the cloud boundary with clear skies, even as the total cloud area remains nearly unchanged. Because the resolution-dependent cloud perimeter is shaped by the complex and chaotic processes of turbulent mixing and diffusion (Hentschel and Procaccia, 1984), and while air and energy exchanges are physically independent of $\xi$, a resolution-based link is required to relate the two (Lovejoy et al., 2001; Fielding et al., 2020).

Fractal geometry is often used as a tool for characterizing the resolution-dependent complexity of shapes. The fractal dimension $D$ was first introduced by Richardson (1961) to characterize the complexity of political borders and was later popularized by Mandelbrot (1967) to describe how the length of a coastline changes depending on the length of the ruler used to measure it. Generally, the perimeter $p$ around an individual fractal object can be related to the measurement resolution $\xi$ through

$$p \propto \xi^{1-D} \tag{1}$$

For the Euclidean case that $p$ is independent of $\xi$ then $D = 1$. At the other extreme, a "space-filling" curve that passes through every resolved point in a unit area has $D = 2$. Lovejoy (1982) first measured $D$ for clouds by relating individual cloud perimeters $p$ to cloud areas $a$ using the expression $p \propto \sqrt{a}^D$. A measured value of $D = 1.35 \pm 0.05 \approx 4/3$ has since been adopted as the canonical value describing individual clouds (Siebesma and Jonker, 2000; Christensen and Driver, 2021), although various studies have shown that $D$ can vary considerably from cloud to cloud. For example, Batista-Tomás et al. (2016) found distinct fractal dimension values for cirrus with ragged, tenuous edges of $D = 1.37$, whereas for cumulonimbus with smoother edges, $D = 1.18$. Other analyses of cumulus fields have found $D = 1.28$ (Zhao and Di Girolamo, 2007) and $D = 1.19$ (Mieslinger et al., 2019) determined using the expression $p \propto \sqrt{a}^D$.

Generally, we define here a geometric quantity that does not vary with length scale as being "scale invariant," such as the scaling of $p$ with $\xi$ in Eq. (1). For such scale invariance to apply to an atmospheric cloud field, this would require that the physics controlling cloud shapes is unchanged with measurement resolution, at least between the limits of possible cloud sizes. Clouds have been shown to be broadly scale invariant for the number distributions of cloud areas and perimeters (DeWitt et al., 2024) despite previous observations of scale breaks that appeared to separate small and large clouds into different physical regimes. DeWitt and Garrett (2024) argue that these scale breaks are artifacts that owe to the treatment of clouds that are truncated by the edge of the measurement domain.

Although the initial result of Lovejoy (1982) showed a constant value of $D$ for length scales ranging from 1 to 1,000 km, suggesting a wide-ranging scale invariance of clouds, the value of $D$ has sometimes been observed to be greater for larger clouds. Cahalan and Joseph (1989) reported $D = 1.27$ for small clouds and $D = 1.56$ for large clouds, supported by Benner and Curry (1998) who found $D = 1.23$ and $D = 1.34$ respectively. Furthermore, after reexamining the data in Lovejoy (1982), Gifford (1989) noted that $D$ increases from 1.35 to 1.77 for the largest clouds with areas $> 2.5 \times 10^4 \text{ km}^2$. The apparent increase in measured $D$ for larger clouds suggests a violation of scale invariance. However, this is likely another artifact of the data

analysis methods. The inclusion of interior cloud holes in area and perimeter measurements has been shown to overestimate calculations of $D$ using the expression $p \propto \sqrt{a}^D$ (Peters et al., 2009; Brinkhoff et al., 2015). Because interior holes tend to fill when imaged with increasingly coarse resolution, this $\xi$ dependence of $a$ results in an inaccurate value of $D$ – the error of which can be calculated using multifractal analysis (Lovejoy and Schertzer, 1991).

Clouds have been shown to be multifractal, such that $D$ is a continuous function of threshold used to distinguish clouds from clear skies (Lovejoy and Schertzer, 1990, 1991; Marshak et al., 1995; Lovejoy and Schertzer, 2006). Studies of the multifractal properties of clouds are useful because they can be used to mathematically account for turbulent intermittency (the variability of turbulent fluctuations), notably observed in measurements of water mixing ratio (Tuck, 2022). We argue that a monofractal assumption is sufficient for the primary conclusions of this study in Section 5.4.

While the fractal dimension and scale invariance are intrinsically linked, their relationship to turbulent structures in the atmosphere is less clear. Two paradigms of turbulence scaling in the atmosphere have been the topic of decades of debate: split 2D and 3D isotropic scaling regimes for large and small scales (Fiedler and Panofsky, 1970; Nastrom et al., 1984), and wide-ranging anisotropic scaling (Lovejoy, 2023). Both theories originated from the pioneering work of Richardson (1926), who showed that the turbulent eddy diffusivity $K$, measured using the relative motion of pairs of particles separated by distance $\ell$, followed a power-law with a 4/3 exponent from the millimeter scale for molecular diffusion to the length scale of atmospheric cyclones ($\ell \sim 10^3$ km), $K \propto \ell^{4/3}$, termed the Richardson "4/3 law" of atmospheric diffusion.

The scaling exponent of the diffusivity with respect to length scale can be obtained experimentally from measurements of velocity fluctuations $\Delta v$ of two air parcels separated by a distance $\ell$ using passive scalars $\Theta$, as a physical quantity that is affected by but does not affect the turbulent flow, such as the concentration of aerosols (Celani et al., 2002). Along one dimension $x$, the generalized first-order (which ignores intermittency) "structure function" expresses the covariance of $\Theta$ as a function of separation distance $\ell$. For turbulent scalars, the function tends to be a power-law given by

$$S(\ell) = |\Delta\Theta(\ell)| = \langle|\Theta(x+\ell) - \Theta(x)|\rangle \propto \ell^{\mathcal{H}} \qquad (2)$$

where brackets indicate averaging over many iterations of the experiment, and $\mathcal{H}$ is the Hurst exponent[1] with bounds $0 < \mathcal{H} < 1$ (Schertzer and Lovejoy, 1984; Hentschel and Procaccia, 1984; Lovejoy and Schertzer, 2012).

The 4/3 law was later derived using dimensional reasoning applied to the theory of 3D isotropic turbulence developed by Kolmogorov (1941). In the theory, for a fluid with kinematic viscosity $\nu$, turbulence kinetic energy is passed along an energy cascade, from large eddies of the energy input scale $L$ to progressively smaller eddies with a constant kinetic energy dissipation rate $\varepsilon$, ending at the "Kolmogorov microscale," $\eta \sim (\nu^3/\varepsilon)^4 \sim 1$ mm, a dissipation length scale where inertial and viscous forces balance. Through dimensional analysis, the covariance of air parcel velocity fluctuations was derived to be $\Delta v \approx \varepsilon^{1/3}\ell^{\mathcal{H}}$, where $\mathcal{H} = 1/3$ for the case of 3D isotropic turbulence. The dimensional approximation that $K \sim \ell v$ (Tennekes and Lumley, 1972) results in $K \sim \varepsilon^{1/3}\ell^{4/3}$, reproducing Richardson's 4/3 power-law, and implying that the relationship between diffusivity

---

[1]The Hurst exponent has various mathematical applications, but here we employ its usage in the field of fractal geometry (for the non-intermittent case) to relate the scaling of turbulent fluctuations with respsect to separation distance $\ell$.

and the Hurst exponent $\mathcal{H}$ (again ignoring intermittency) follows

$$K \sim \ell^{1+\mathcal{H}} \tag{3}$$

As Sect. 5 elaborates, the value of $\mathcal{H}$ differs based on the dimensionality of the turbulence (e.g., the case of 2D isotropic turbulence). The problem that 3D turbulence cannot apply at the "flatter" planetary scales to a relatively thin troposphere has been well known. Even Kolmogorov predicted that 3D turbulence can only apply in the atmosphere at scales < 100 m. This led to the paradigm that 3D isotropic turbulence must be applicable at small scales and 2D at large scales, separated by a

95 scale break around the depth of the troposphere (See Lovejoy (2023) for a historical review.). The contrasting case for 2D turbulence was developed for the case of an incompressible fluid (Kraichnan, 1967) and later refined into the theory of 2D quasi-geostrophic turbulence (Charney, 1971), where the expected value is instead $\mathcal{H} = 1$.

Figure 1 illustrates how two- and three-dimensional components in cloud structures are visible at all scales, but arguably 2D structures predominate at scale $L$, becoming more 3D approaching $\eta$, reflecting a scale dependence due to large-scale

stratification. Aircraft measurements of turbulent spectra of wind and temperature fluctuations have been argued to support this physical separation of large and small scales, where quasi-two-dimensional structures are seen at large scales and isotropic three-dimensional structures at small scales (Fiedler and Panofsky, 1970), with a scale break seen between approximately 20 km and 500 km (Nastrom et al., 1984; Gage and Nastrom, 1986). Lovejoy et al. (2009) argued that this scale break is an artifact owing to vertical aircraft movements that occur when flying along isobars rather than isoheights and proposed instead that 3D

isotropic turbulence is inapplicable at nearly any scale because stratification compresses the atmosphere vertically, even for scales as small as 5 m. Furthermore, the results of Alder and Wainwright (1970) show the formation of vortices even at the $10^{-8}$ m scale, inconsistent with a description of isotropic molecular diffusion (Tuck, 2022).

Specifically, Lovejoy et al. (2007) (hereafter L07), and more comprehensively Lovejoy and Schertzer (2013), provided evidence that, rather than two separate isotropic turbulence regimes, the atmosphere is best characterized by a single anisotropic

turbulence regime spanning all scales in the atmosphere. Following the framework of generalized scale invariance (GSI), which accounts for stratification, the "23/9D" elliptical model of turbulence in the atmosphere is characterized by a dimension intermediate to 2D and 3D (Schertzer and Lovejoy, 1985). Power spectra of radar reflectivity, cloud radiance, wind speed, and temperature all revealed length-scaling exponents that lie between purely 2D and 3D turbulence cases, consistent with an anisotropic turbulence regime predicted to have a volume dimension of $D = 2.55 = 2 + \mathcal{H}_z$ where $\mathcal{H}_z \approx 0.55$ is the ratio of

horizontal and vertical values of $\mathcal{H}$ (discussed further in Sect. 5) (Schertzer and Lovejoy, 1985; Lovejoy and Schertzer, 1985; Lovejoy et al., 1993; Lovejoy, 2021). For the Gaussian case, which does not include intermittency or multifractal aspects, $\mathcal{H}$ is calculated from the power spectrum of the observed phenomenon, $E(k) \sim k^{-B}$, where $B = 2\mathcal{H} + 1$. In the 23/9D theory, which incorporates the vertical and horizontal aspects of separation, $\mathcal{H}_z = (B_V - 1)/(B_H - 1)$.

Simplifications of the first-order structure function have also been used to determine $\mathcal{H}$ for properties of clouds (Pressel and

120 Collins, 2012; Pressel et al., 2014), and to link the dimension of turbulence to the fractal dimension through the expression (Hentschel and Procaccia, 1984; Mandelbrot, 1985)

$$D = 2 - \mathcal{H} \tag{4}$$

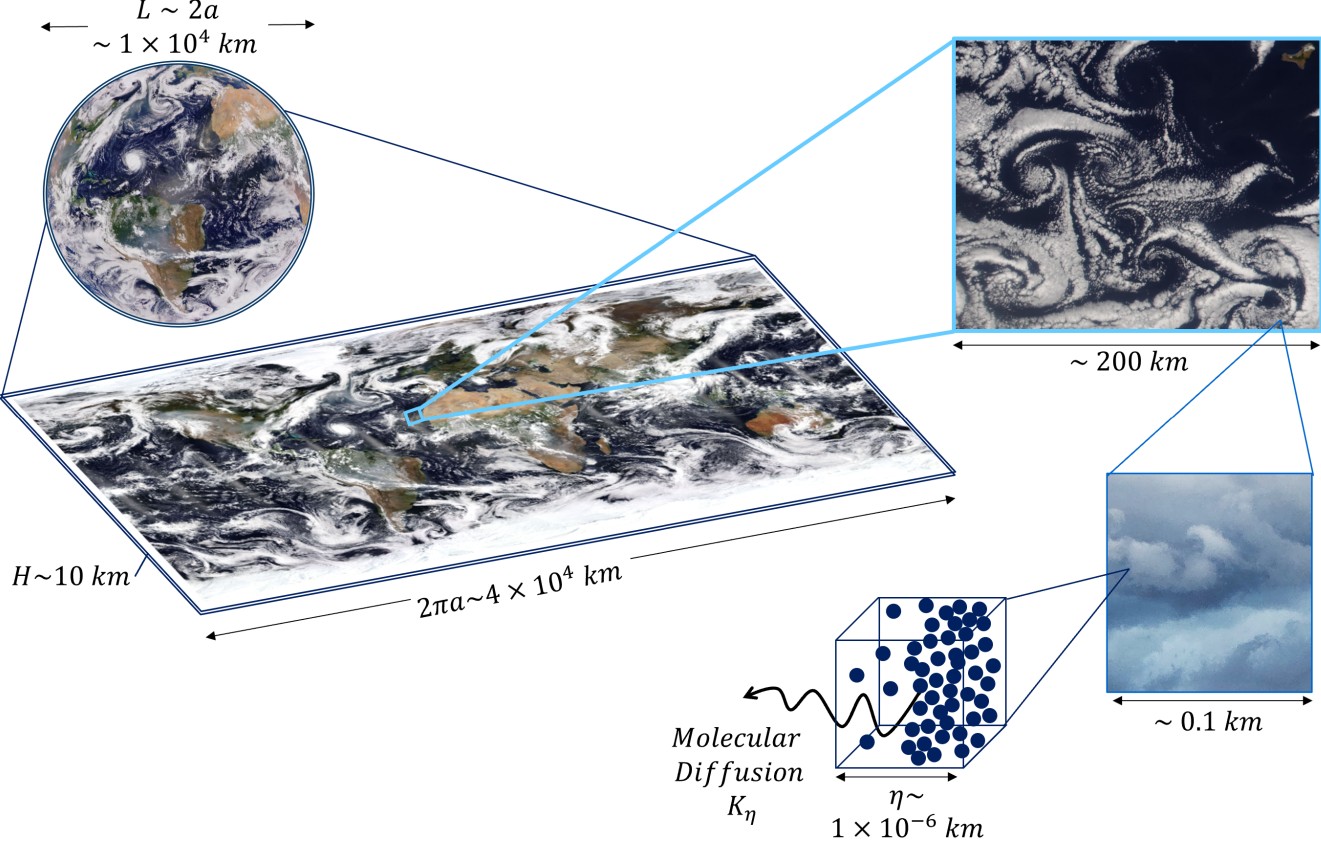

**Figure 1.** Diagram showing the similarity of rotational motions of clouds from the planetary diameter $L \sim 2a$, where structures are nearly 2D, to smaller scales with more 3D structure. At $L$ (left), swirling features associated with synoptic-scale systems are $\sim 10^4$ km long and nearly 2D compared to the tropospheric depth $H$. At smaller resolved scales, the vertical component is more similar to the horizontal component, and thus the structure is more 3D. The images on the left are from EPIC (top) and a MODIS and VIIRS composite (bottom) for the same time frame. The upper right inset shows cloud features shaped by von Kármán vortices viewed near the Canary Islands obtained by VIIRS with eddy length scales $\sim 10$ km. The image on the bottom right shows swirling clouds in a thunderstorm photographed from the ground with a length scale $\sim 0.01$ km. The bottom inset is a cartoon depicting the smallest length scale of turbulence, the Kolmogorov microscale $\eta$, where kinetic energy is dissipated to heat through molecular diffusion $K_\eta$, with individual cloud droplets illustrated as dots with spacing to represent the cloud edge interface.

Equation (4) is the 2D analog of the fractal dimension of a geometric set of points. For example, given $(x, \Theta(x))$ where $x$ is the position in a 1D transect and $\Theta$ is the measured cloud brightness, the 1D case $D = 1 - \mathcal{H}$ extends to the 2D cloud perimeter $(\Theta(x, y)$ as $D = 2 - \mathcal{H}$ (Hentschel and Procaccia, 1984).

Observations of scaling behaviors in clouds, whether expressed through the fractal dimension or turbulent structure functions, point to a robust relationship between $\xi$, cloud geometry, and turbulence. This paper explores the topic as follows. In Sect. 2, we relate the Hurst exponent to an "ensemble" fractal dimension $D_e$ that defines a globally distributed cloud field and discuss in Sect. 3 a resolution coarsening procedure to measure it. Section 4 presents the values of the ensemble fractal dimension obtained using several satellite and numerical model datasets. Section 5 interprets the significance of the results by comparing them to the expected values of $D_e$ and $\mathcal{H}$ for 2D and 3D isotropic turbulence, as well as for an anisotropic turbulence regime that is intermediate to 2D and 3D at all scales. Our findings contradict the theories proposing split 2D and 3D isotropic turbulence regimes separated by a scale break that have prevailed over the past decades (Fiedler and Panofsky, 1970; Nastrom et al., 1984), and support the concept of a wide-ranging, scale invariant 2D-3D anisotropic turbulence regime proposed by Schertzer and Lovejoy (1985), described in detail by Lovejoy and Schertzer (2013). We show that this anisotropic turbulence regime applies to cloud perimeters over a remarkable 10 orders of magnitude ranging from the Kolmogorov microscale $\eta$ to the planetary diameter $2a$.

## 2 Analytical expressions relating the perimeter of cloud ensembles to the dimension of turbulence

To explore how cloud perimeter varies with measurement resolution $\xi$, the total perimeter of a cloud ensemble viewed from above (e.g., looking down as a satellite would view it from space) can be expressed in terms of a "perimeter density." The perimeter density $\mathcal{P}$ is defined as the summed perimeters $p$ of all clouds $P = \sum p$ normalized by the area of the horizontal domain $A_d$; that is, $\mathcal{P} = P/A_d$, a quantity analogous to the cloud fraction $\mathcal{A} = A/A_d$ where $A$ is the total cloud area. In this section, we show how $\mathcal{P}$ can be related to $\xi$ through the Hurst exponent $\mathcal{H}$.

### 2.1 Cloud perimeter and the Hurst exponent

In Garrett et al. (2018), the total cloud edge perimeter $P$ of a tropical convective cloud field was estimated theoretically for equal horizontal and vertical resolutions $\xi_H = \xi_V = \xi$ within a domain volume $V = A_d \xi_V$. To obtain $P$, a "mixing engine" framework was introduced, that described cloud edge circulations consisting of coupled large-scale vertical buoyancy oscillations and horizontal turbulent exchanges as shown in Fig. 2. The derivation reflects a dimensional balance between two speeds. In the horizontal, $v_H = K\mathcal{P}$ represents a speed of erosion or formation of cloud edge due to dissipative mixing with a characteristic length scale $\xi_H$. The speed in the vertical direction is $v_V = \mathcal{N}\xi_V$ where $\mathcal{N}$ is the moist adiabatic Brunt-Väisälä frequency, and represents the speed of production of potential energy through oscillatory vertical motions. Assuming steady-state and that the speeds of the horizontal and vertical legs of the circulation are equal, $v_V = v_H$, then it follows that

$$K\mathcal{P} = \mathcal{N}\xi_V \tag{5}$$

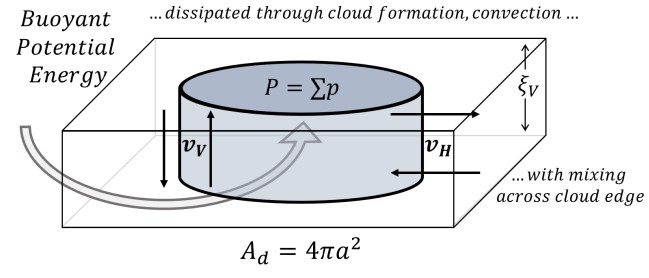

**Figure 2.** Illustration of the theorized cloud edge mixing engine from Garrett et al. (2018). The circulations are generated from the production and dissipation of buoyant potential energy at the planetary scale. All clouds in the domain area $A_d$ of Earth's surface are represented as a single cylinder with total perimeter $P = \sum p$. The total available buoyant potential energy is dissipated vertically through moist adiabatic convection with vertical buoyancy speed $v_V \sim \mathcal{N}\xi_V$, and horizontally via turbulent mixing at cloud edge with speed $v_H \sim KP$. Globally, the vertical and horizontal components must balance.

Invoking mass continuity for the cloud edge circulation, $\nabla \cdot \mathbf{v} = 0$ leads to $\partial v_V / \partial \xi_V = -\partial v_H / \partial \xi_H$, and through scale analysis, $\mathcal{N}\xi_V / \xi_V \sim K\mathcal{P} / \xi_H$. Thus, where $\xi_H$ is the horizontal measurement resolution $\xi$ viewed from space,

$$\mathcal{P}_\xi \sim \frac{\mathcal{N}\xi}{K_\xi} \tag{6}$$

where $K_\xi$ is the turbulent eddy diffusivity with eddy length scale $\xi$.

From Eq. (3), $K_\xi$ scales with $\mathcal{H}$, the value of which varies depending on the dimensionality of turbulence reflecting any anisotropy between $\xi_V$ and $\xi_H$. The adjustment needed to scale $K_\xi$ from the molecular diffusivity at $\eta$ (i.e., $K_\eta$, the diffusion coefficient of air) to the resolution $\xi$ is (Richardson, 1926; Garrett et al., 2018)[2]

$$K_\xi = K_\eta \left(\frac{\xi}{\eta}\right)^{1+\mathcal{H}} \tag{7}$$

Substituting Eq. (7) into Eq. (6), the expected relationship relating measurement resolution to the cloud perimeter density $\mathcal{P}_\xi$ becomes:

$$\mathcal{P}_\xi = \frac{\mathcal{N}\eta}{K_\eta} \left(\frac{\eta}{\xi}\right)^{\mathcal{H}} \propto \xi^{-\mathcal{H}} \tag{8}$$

An observed value of $\mathcal{H}$ is then obtainable from measurements of $\mathcal{P}$ as a function of $\xi$.

## 2.2 The fractal dimension of cloud ensembles

Equation (8) expresses the cloud perimeter as a function of resolution, and is thus analogous to Eq. (1) where $p \propto \xi^{1-D}$ with fractal dimension $D$. The canonical value for individual clouds is $D \approx 4/3$ (Lovejoy, 1982; Siebesma and Jonker, 2000; Chris-

---

[2]Note that $\xi$ is normalized here by $\eta$ rather than the more common normalization by outer scale $L$, the largest eddy of the turbulent flow, from which energy is transferred to smaller eddies of observation scale $\ell = \xi$ in the energy cascade. Because the choice of normalization length scale does not affect calculations of the value of $\mathcal{H}$ or $D_e$, we choose $\eta$ to relate $\mathcal{P}_\xi$ to $K_\xi$ and $K_\eta$. This is consistent with the approach taken by Krueger et al. (1997); Garrett et al. (2018) who focused on the relationship between cloud measurements at scale $\xi$ and turbulent processes at the Kolmogorov microscale $\eta$.

tensen and Driver, 2021), but there are complications with this expression for $D$, including the aforementioned multifractal nature of clouds. Additionally, $D$ in the expression $p \propto \sqrt{a}^D$ only applies mathematically to the shape of an individual cloud, and has been argued to only represent sets of identically shaped objects (Imre, 1992).

From a climatological perspective, it is instead the ensemble of clouds with total perimeter density $\mathcal{P}$ that governs exchanges of energy and air across cloud edges. Following the fractal "islands" analogy from Mandelbrot (1977) who considered the total perimeter of an ensemble of objects (described in more detail below), we propose an "ensemble fractal dimension" for clouds $D_e$ analogous to Eq. (1) such that

$$\mathcal{P}_\xi \propto \xi^{1-D_e} \tag{9}$$

implying from Eq. (6), that the scaling exponent of the diffusivity is equivalent to the ensemble fractal dimension:

$$K_\xi \propto \xi^{D_e} \tag{10}$$

The distinction between $D$ and $D_e$ was first raised by Mandelbrot (1977), who showed that an ensemble of fractal "islands" with power-law distributed areas $a$ follows the Korčák Law, a survival function $S(a' > a) \propto a^{-\mathcal{K}}$ (Korčák, 1938), where the ensemble fractal dimension of the total coastline perimeter $P$ is $D_e = 2\mathcal{K}$ confined to the bounds $1 \le D < D_e \le 2$.

The survival function can be related to a cumulative distribution function (CDF) through $S = 1 - CDF$ and $\mathcal{K}$ is equivalent to the exponent of the power-law number distribution (Clauset et al., 2009). The area number distribution can be expressed as $n_a \propto a^{-(1+\alpha)}$ for clouds (Cahalan and Joseph, 1989; Benner and Curry, 1998; Wood and Field, 2011), and $\mathcal{K} \sim \alpha$. The perimeter number distribution, $n_p \propto p^{-(1+\beta)}$, is related to that for the area through $\alpha = D\beta/2$ for clouds (DeWitt et al., 2024). It follows that the ensemble fractal dimension is given by $D_e = D\beta$. The inequality $D < D_e$ requires that $\beta > 1$.

Comparing the exponents in Eqs. (8) and (9), the Hurst exponent can be related to the ensemble fractal dimension through

$$\mathcal{H} = D_e - 1 \tag{11}$$

This equation is important because it provides a means for linking satellite observations of cloud perimeter fractal properties and size distributions to the less easily seen but more physically relevant turbulent structures at cloud edge. For comparison with a LES model of a tropical cloud field resolved at 100 m scales, Garrett et al. (2018) applied a value of $\mathcal{H} = 1/3$ to Eq. (7) consistent with Richardson (1926) and the 4/3 law. Implicit in this case is an assumption of 3D isotropic turbulence at resolved scales. The assumption may be appropriate for an LES that chooses a cubic Eulerian grid for computational ease at the expense of losing a Lagrangian perspective.

However, while $D = 4/3$ is consistent with values seen for individual clouds, a larger value is required for cloud ensembles, in which case the inequality $D < D_e$ predicted by Mandelbrot (1977); DeWitt et al. (2024) applies. In a similar adjustment to the individual fractal dimension, Hentschel and Procaccia (1984) related the perimeter fractal dimension of clouds to $\mathcal{H}$ through the expression $D = 2 - \mathcal{H}$ (Eq. 4), with a correction for turbulent intermittency ($\mu$, where $D_\mu = (4+\mu)/3 \approx 5/3$ (described below). We obtain, from Eqs. (11) and (4), an adjustment to $D$ for an ensemble of clouds:

$$D_e = 3 - D \tag{12}$$

**Table 1.** Summary of main formulas

| Equation number | Formula | Reference |
|:---:|:---:|:---|
| (1) | $p \propto \xi^{1-D}$ | Mandelbrot (1967) |
| (2) | $S(\ell) = \Delta\Theta(\ell) = \langle \Theta(x+\ell) - \Theta(x) \rangle \propto \ell^{\mathcal{H}}$ | Kolmogorov (1941) |
| (3) | $K \sim \ell^{1+\mathcal{H}}$ | Derived from Richardson (1926) and Kolmogorov (1941) |
| (4) | $D = 2 - \mathcal{H}$ | Hentschel and Procaccia (1984) |
| (5) | $K\mathcal{P} = \mathcal{N}\xi_V$ | Garrett et al. (2018) |
| (6) | $\mathcal{P}_\xi \sim \frac{\mathcal{N}\xi}{K_\xi}$ | Equation (5) |
| (7) | $K_\xi = K_\eta \left(\frac{\xi}{\eta}\right)^{1+\mathcal{H}}$ | Krueger et al. (1997) and Eq. (3) |
| (8) | $\mathcal{P}_\xi = \frac{\mathcal{N}\eta}{K_\eta} \left(\frac{\eta}{\xi}\right)^{\mathcal{H}} \propto \xi^{-\mathcal{H}}$ | Equations (6) and (7) |
| (9) | $\mathcal{P}_\xi \propto \xi^{1-D_e}$ | Mandelbrot (1977) |
| (10) | $K_\xi \propto \xi^{D_e}$ | Equations (6) and (9) |
| (11) | $\mathcal{H} = D_e - 1$ | Equations (8) and (9) |
| (12) | $D_e = 3 - D$ | Equations (4) and (11) |

The quantity $3 - D$ has been defined as the intermittency exponent by Hentschel and Procaccia (1984) and the multifractal codimension[3] (Schertzer and Lovejoy, 1987) within a 3D space. Applying the canonical value of $D = 4/3$ for individual clouds leads to the expected value of $D_e = 5/3$ for cloud ensembles. Perhaps the geometric intermittency of multiple and varied cloud types in an ensemble reflects the turbulent intermittency that is not represented by $D$ for individual clouds. This estimate of $D_e = 5/3$ is in agreement with Hentschel and Procaccia (1984), who found that Richardson's 4/3 law only applies if the fractal dimension is $D_\mu = 5/3$, obtained by adding an intermittency correction with a value between $0.25 < \mu < 0.5$ to the value $D = 4/3$. The 5/3 value is also nearly identical to the value of $D_e = 1.68 \pm 0.06$ obtained from $D_e = D\beta$ for $\beta = 1.26$ from DeWitt et al. (2024), which applied across various satellite instruments and climate regimes. In this case, the implied value of the Hurst exponent is $\mathcal{H} = 2/3$, which is between the 3D turbulence value of $\mathcal{H} = 1/3$ and the 2D turbulence value of $\mathcal{H} = 1$. Using Eq. (9) and the methods below, we observationally evaluate the applicability of the result $\mathcal{H} = 2/3$ that is associated with $D_e = 5/3$.

## 3 Data and methods

Equation (11) implies that the dimensionality of the turbulent structure in clouds can be inferred from observations of cloud perimeters. To explore this hypothesis and the suggestion from Eq. (12) that $\mathcal{H} = 2/3$, we consider satellite imagery of clouds from platforms in polar-orbiting, geostationary, and heliocentric orbits (summarized in Table 2) using cloud mask algorithms. The resulting binary arrays, hereafter "cloud masks," represent cloudy pixels with a value of unity and clear skies by a value of zero. Cloudy pixels are considered an individual connected cloud when they are vertically or horizontally adjacent (i.e.,

---

[3]The difference between the spatial dimension of the domain and the fractal dimension

"4-connectivity"). The edges of the domain are not included as part of the perimeter. The quantities $p$ and $a$ are calculated for all individual clouds (see Fig. 3f for an example). The perimeter is defined as the sum of all pixel edge lengths along the outer

edge of each cloud. Although the example shows that all pixel sizes are equal, in satellite imagery, each pixel has individual values of $\xi_x$ and $\xi_y$ for its width and height, which are adjusted from $\xi_N$ to account for the Earth's curvature away from the satellite nadir vertically and horizontally. The area is the sum of $\xi_x \times \xi_y$ for each pixel in the cloud. For each image, $p$ and $a$ are summed and normalized by domain area $A_d$ to determine $\mathcal{P}$ and $\mathcal{A}$.

The polar-orbiting data sets considered are from the instruments VIIRS and MODIS, which have native pixel resolutions

$\xi_N$ at the nadir of 0.75 km and 1 km, and capture imagery in narrow, meridional swaths. Values of $\xi_N$ represent the pixel resolution at satellite nadir; the horizontal and vertical dimensions of each pixel are adjusted based on their distance from the nadir. The average swath size for VIIRS is 2501 × 12944 pixels with a domain area $A_d$ of $2.6 \times 10^7$ km$^2$. For MODIS it is 1261 × 8120 pixels with $A_d = 2.0 \times 10^7$ km$^2$. The VIIRS and MODIS datasets include 60 and 72 cloud masks from 02 June 2021. Their respective cloud mask techniques are described by Kopp et al. (2014) and Ackerman et al. (2008). We also include

12 MODIS cloud masks with 0.25 km resolution as defined by optical reflectance values $R \geq 0.01$ described by DeWitt et al. (2024). These high-resolution cloud masks have $A_d = 5.1 \times 10^6$ km$^2$ and average image dimensions of 5048 × 8120 pixels obtained from 01 January 2021 through 09 January 2021.

Geostationary datasets are obtained from instruments denoted here by their more familiar satellite names, Himawari (instrument name: AHI), GOES-WEST (ABI), and METEOSAT-11 (SEVIRI), which provide full-disk imagery of Earth with

$A_d = 1.0 \times 10^8$ km$^2$ positioned over the fixed longitudes 141°E, 137°W, and 0°, respectively. Their nadir pixel resolutions are 2 km, 2 km, and 3 km, respectively, with cloud masks as described by Derrien and Gléau (2005). For each of the geostationary datasets, 30 cloud masks were obtained from 02 June 2021 through 01 July 2021, each at approximately the local noon at the satellite nadir.

To provide unique observations of global cloud coverage, we also include cloud masks from GEO-Ring and EPIC. GEO-

Ring is a composite of geostationary satellite imagery (Ceamanos et al., 2021) that provides stitched satellite imagery of the surface of the Earth (excluding the poles) with $A_d = 4.4 \times 10^8$ km$^2$ at $\xi_N = 11$ km. Thirty-nine GEO-Ring cloud masks were obtained from 02 June 2021 through 21 June 2021. EPIC obtains full-disk imagery of Earth from the DSCOVR satellite in heliocentric orbit, photographing Earth as it rotates, providing coverage of all longitudes. Due to its location at the L1 Lagrange Point in deep space, EPIC imagery has a coarser pixel resolution of 8 km. 30 EPIC cloud masks (described by described by

Yang et al. (2019)) were obtained from 01 June 2017 through 30 June 2017.

As a means to compare measurements of $\mathcal{P}_\xi$ from satellite observations to the value derived by Garrett et al. (2018), we consider the geometries of clouds simulated using the System for Atmospheric Modeling (SAM), calculated as they would be viewed from space. SAM is a high-resolution 3D LES, initialized with idealized GATE Phase III campaign soundings for tropical convection. The simulation domain of 204.8 km × 204.8 km × 19 km includes more than one billion grid points –

often referred to as a "Giga-LES." There are 2048 grid points in the horizontal directions with a grid spacing of 100 m, and 256 grid points in the vertical with grid spacing ranging from 50 m to 100 m. The simulation is integrated at two-second intervals

**Table 2.** Summary of satellite datasets used in this study.

| Dataset name | Sensor name | View Type | Approx. nadir resolution | Longitude at nadir | Dates examined | Description of cloud mask algorithm |
|---|---|---|---|---|---|---|
| **MODIS 250 m** | MODIS | Polar-Orbiting | 250 m | - | 01 January 2021 to 09 January 2021 | DeWitt et al. (2024) |
| **VIIRS** | VIIRS | Polar-Orbiting | 750 m | - | 03 June 2021 to 04 June 2021 | Kopp et al. (2014) |
| **MODIS 1km** | MODIS | Polar-Orbiting | 1 km | - | 02 June 2012 to 02 June 2012 | Ackerman et al. (1998, 2008) |
| **Himawari** | AHI | Full-Disk | 2 km | 141° E | 02 June 2021 to 01 July 2021 | Derrien and Gléau (2005, 2010) |
| **GOES** | ABI | Full-Disk | 2 km | 137° W | 02 June 2021 to 01 July 2021 | Derrien and Gléau (2005, 2010) |
| **METEOSAT** | SEVIRI | Full-Disk | 3 km | 0° | 02 June 2021 to 01 July 2021 | Derrien and Gléau (2005, 2010) |
| **EPIC** | EPIC | Full-Disk | 8 km | - | 01 January 2017 to 31 January 2017 | Yang et al. (2019) |
| **GeoRing** | (Composite) | Full-Disk | 11 km | - | 02 June 2021 to 21 June 2021 | Ceamanos et al. (2021) |

for 24 hours. Refer to Khairoutdinov et al. (2009) for a more complete description of the simulation. We analyze scenes hourly from hour 12 through hour 24 of the model to ensure that steady-state has been reached.

In order to compare the 3D model data with 2D satellite retrievals, we define the SAM cloud masks as 3D binary arrays
for mixing ratios of non-precipitating cloud condensate $q_n > 0.01$ g kg$^{-1}$. A 2D facsimile of a satellite cloud mask is created from a vertical projection of the 3D cloud mask that represents the view from above. The 2D binary cloud mask is assigned a cloudy pixel based on threshold value $j$ where the corresponding vertical columns of the 3D cloud mask have $\sum_0^H$ (cloudy pixels) $> j$. For example, with a threshold value of $j = 3$, each pixel in the 2D cloud mask is classified as cloudy if more than three of the pixels in the corresponding 3D vertical column are cloudy. Multiple 2D cloud masks were obtained using
threshold values $j = 1, 2, 3, 5, 9, 15$. This thresholding procedure is similar to an analysis performed by DeWitt et al. (2024) that compared differences in cloud statistics defined by various optical depth thresholds.

To obtain values of $D_e$, total cloud perimeter $P$ is calculated first at the native spatial resolution $\xi_N$ and normalized by $A_d$ to obtain $\mathcal{P}_N$. The image is then artificially coarsened (see description below) and the procedure is repeated. $\mathcal{P}_\xi$ is obtained at progressively coarser spatial resolutions $\xi > \xi_N$ such that $\xi = \xi_N k$, where $k$ is the coarsening factor. Coarsening is performed
by separating the original image into a grid of multiple "boxes" containing $k \times k$ pixels (see Fig. 3e, red boxes) which are reduced to a single upscaled pixel through averaging. Each pixel of the coarsened image (Fig. 3f, outlined in blue) is determined

to be cloudy or clear by rounding the average of the values inside each box in the native resolution image (Fig. 3e, outlined in blue) to unity or zero. The values of $k$ are chosen to be the nearest odd integers that differ by a constant factor (e.g., $k = 2^n$). The maximum value of $k$ for each dataset corresponds to coarsening to a single pixel along the shorter dimension of the domain. Figure 3 (a-d) shows an example of resolution coarsening to a single pixel for various EPIC cloud masks.

A least squares linear regression is performed on values of $\ln \mathcal{P}_\xi$ and $\ln \xi$ to obtain the Hurst exponent $\mathcal{H}$ and $D_e$ from Eqs. (8) and (9). Linear regression was performed on the straightest region of all curves, which was found to be $7 < \xi/\xi_N < 150$ where biases due to interpolation ($\xi/\xi_N < 7$, most significantly for EPIC) and due to the square shape of pixels at very coarse resolutions ($\xi/\xi_N > 150$) are omitted. Uncertainties in the linear regression are evaluated at the 95% confidence level.

## 4 Results: Cloud measurements

Alongside measurements of perimeter density $\mathcal{P}_\xi$, the more familiar quantity of cloud fraction $\mathcal{A}_\xi$ is included as a point of comparison. Figure 4 shows the cloud fraction and perimeter density obtained from satellite and model datasets at native resolution $\xi_N$, termed $\mathcal{A}_N$ and $\mathcal{P}_N$, and coarsened resolutions $\xi$ normalized by $\xi_N$. Both $\xi_N$ and the domain areas $A_d$ span two orders of magnitude: $\xi_N$ from 0.1 km to 11 km, and $A_d$ from $5.1 \times 10^6$ to $4.4 \times 10^8$ km$^2$.

### 4.1 Measured cloud fraction $\mathcal{A}$

Global cloud fraction values $\mathcal{A}_N$ in Fig. 4a range from 0.5 to 0.7, reflecting differences in cloud mask techniques. With progressive coarsening, $\mathcal{A}$ changes by less than 5% before bifurcating at $\xi/\xi_N \sim 100$. As $\xi$ approaches $\xi/\xi_N \sim 1,000$, $\mathcal{A}$ is represented by a single pixel, with a value of either zero or unity (except for the polar-orbiting satellites, which are represented by a $1 \times 5$ line). Interestingly, geostationary cloud fraction measurements with $\mathcal{A}_N > 0.56$ approach a value of unity, whereas MODIS 0.25 km and EPIC datasets with $\mathcal{A}_N = 0.55$ instead trend toward zero.

This bifurcation of cloud fraction reflects that as an image of a cloud field is coarsened to a single pixel, the coarsened pixel value is determined by averaging and rounding to zero or unity the pixel values in the original image (illustrated in Sect. 3). Conversely, a coarsening method in which the presence of any cloudy pixel in the original image results in a cloudy coarsened pixel causes cloud fraction to converge to unity with coarsening (Di Girolamo and Davies, 1997). Due to this averaging method, the value to which cloud fraction bifurcates depends on the native cloud fraction $\mathcal{A}_N$. Figure 3 shows four examples of coarsening EPIC cloud masks to a single pixel, resulting in either a single clear or cloudy pixel. Although statistically the initial cloud fraction value is a good indicator of whether the single pixel will be cloudy or clear, it is not the only factor. For $\mathcal{A}_N = 0.55$, the single-pixel value of $\mathcal{A}$ depends more on the initial *distribution* of clouds. Where clouds are more evenly distributed across the globe (Fig. 3b), smaller isolated structures vanish more quickly with coarsening and approach $\mathcal{A} = 0$ for a single pixel. When clouds are more clustered (Fig. 3c), a coarsened single pixel has $\mathcal{A} = 1$.

The application of vertical pixel thresholding in SAM results in a wide range of native cloud fraction values between $0.15 < \mathcal{A}_N < 0.60$. Larger threshold values $j$ tend to exclude small and shallow clouds, and in turn, decrease the overall cloud fraction. Bifurcation of $\mathcal{A}$ occurs for SAM at a native value of $\mathcal{A}_N \approx 0.45$, notably smaller than the value at which bifurcation

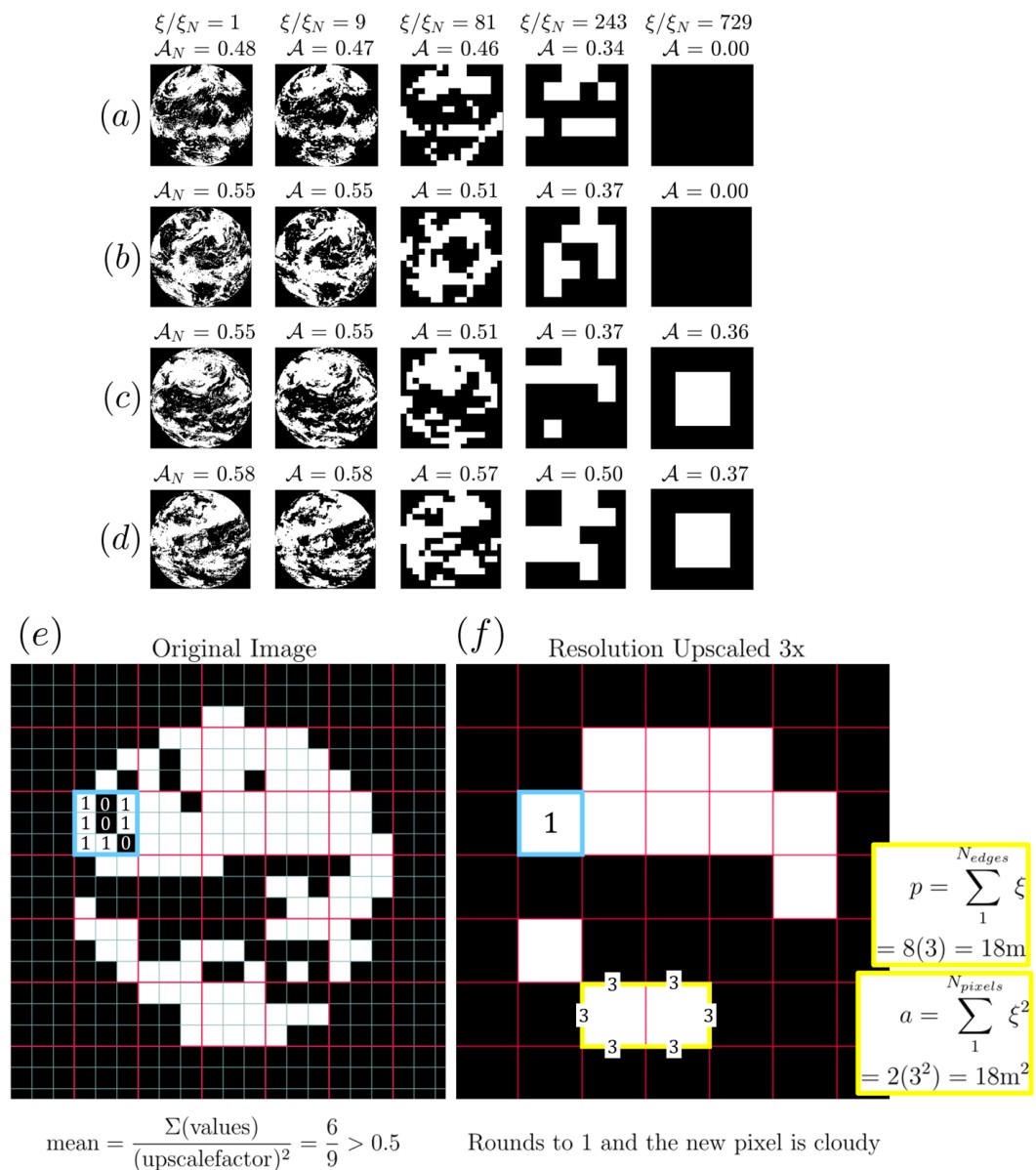

**Figure 3.** Top: EPIC cloud masks shown at native resolution $\xi_N$ and coarsened resolutions $\xi$ to a single pixel for four cases with initial native cloud fraction between $0.48 < \mathcal{A}_N < 0.58$ (increasing from top to bottom) illustrating a bifurcation of cloud fraction with coarsening of resolution depending on the native cloud fraction to either zero or unity. Note that the single pixel case shown here has a value of $\mathcal{A} = 0.37$ rather than unity because the domain area represented by the square pixel is the disk area $A_d = \pi a^2$. Bottom: A detailed example of the upscaling method. The "original" image here is shown at 100x the resolution of the true original image to exaggerate pixels for clarity, and the coarsened image is upscaled 3x or $k = \xi/\xi_N = 3$. The thin blue lines outline pixels with side lengths $\xi_N$ and the red boxes are the upscaled pixel regions. An example region of pixels from which the mean is used to determine whether the upscaled pixel is cloudy or clear is highlighted as a thick blue box with pixel values shown within. An example of the individual perimeter and area calculation is highlighted in yellow, assuming the original pixel resolution is $\xi_N = 1$ m. Areas outside the circle are marked NaN and so are omitted from the average.

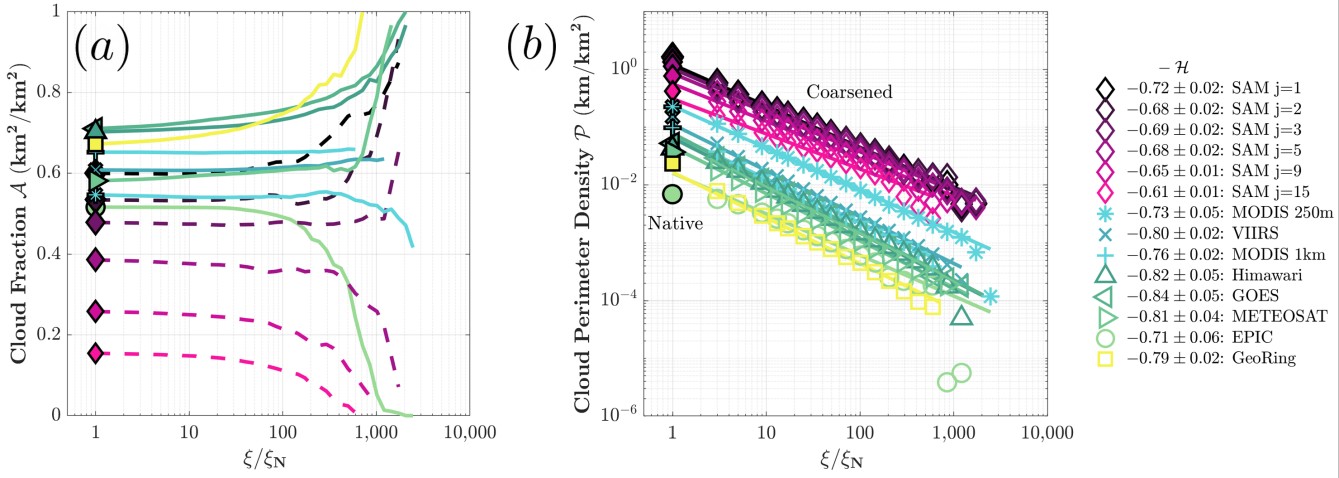

**Figure 4.** Measured cloud fraction $\mathcal{A}_N$ (a) and perimeter density $\mathcal{P}_N$ (b) at native measurement resolution $\xi_N$ (solid markers), and $\mathcal{A}_\xi$ and $\mathcal{P}_\xi$ at coarsened resolutions $\xi$ normalized by $\xi_N$ (lines and hollow markers) for polar-orbiting (blue), full-disk (green), and global mosaic (yellow) satellite datasets. The SAM numerical simulations are shown as pink diamonds (with brightness scaled by threshold value $j$). Legend entries are sorted by increasing $\xi_N$ with the associated negative Hurst exponent $\mathcal{H}$ obtained from a least squares linear regression of $\ln \mathcal{P}_\xi$ and $\ln \xi$ (Eq. 8) and uncertainties evaluated for a 95% confidence interval. Linear regression was performed on the straightest portion of all curves, found to be $7 < \xi/\xi_N < 150$, although the best fit lines are extrapolated to all points to show their relative distance to the fit.

occurs for satellite datasets at $\mathcal{A}_N \approx 0.55$. This discrepancy suggests a difference between the clustering behavior of clouds
viewed globally by satellite and those of modeled clouds for a region of tropical convection. A possible explanation for the discrepancy is that models assume local thermodynamic equilibrium, which has been argued not to apply in the atmosphere (Tuck, 2022).

### 4.2 Measured cloud perimeter density $\mathcal{P}$

The resolution dependence of cloud perimeter density $\mathcal{P}$ can be defined more simply than for cloud fraction $\mathcal{A}$. As shown in
Fig. 4b, perimeter density $\mathcal{P}_\xi$ has a power-law scaling with $\xi$ in all datasets, independent of satellite orbit, domain size, and resolution. For $\xi > \xi_N$, $\mathcal{P}_\xi$ is well characterized by a linear regression of $\ln \mathcal{P}_\xi$ to $\ln \xi$ (Eq. 8). Linear regression is performed only on data points in the straightest region of all curves, $7 < \xi/\xi_N < 150$, where biases due to interpolation ($\xi/\xi_N < 7$) and the square shape of pixels at very coarse resolutions ($\xi/\xi_N > 150$) are ignored. The lines in the figure are shown extrapolated to all points to demonstrate their relative distance to the least-squares fit, which can be assumed to more reliably reflect the
physical fractal nature of the cloud ensemble. This power-law relationship $\mathcal{P}_\xi \propto \xi^{-\mathcal{H}}$ holds even past the point $\xi/\xi_N \sim 100$ where cloud fraction values $\mathcal{A}$ tend to diverge. However, for very large $\xi/\xi_N \sim 1{,}000$, $\mathcal{P}_\xi$ can deviate from the power-law regression to lower values, reflecting the fractal nature of the problem: complex cloud structures cannot be fully represented by coarse Euclidean geometries such as a single square pixel. This low bias in $\mathcal{P}_\xi$ for values of $\xi/\xi_N$ between $\sim 100$ and $\sim 1000$

can also be seen in the fourth and fifth columns of Fig. 3. There, for $\xi/\xi_N = 243$ the images appear pixelated, but maintain their general structure. However, for $\xi/\xi_N = 729$, the cloud mask consists of either a single cloudy or a clear pixel. The value of $\xi/\xi_N$ at which $\mathcal{P}$ begins to depart from the linear regression corresponds to the coarsest resolution for which the complexity of the cloud edge can still be reliably measured.

Notably, the value of $\mathcal{P}_N$ for the native resolution $\xi/\xi_N = 1$ does not always align with the best-fit line, especially for the case of EPIC. As discussed in DeWitt et al. (2024), EPIC employs an on-board averaging and post-processing interpolation that artificially smooths the edges of clouds to compress data for transmission. This interpolation results in the artifact that $\mathcal{P}_N$ is lowered due to the reduced edge complexity. A similar phenomenon is observed to a lesser degree for the other satellite datasets. To avoid this issue, the $\mathcal{P}_N$ data points are not included in the linear regression.

For satellite datasets, values of $\mathcal{H}$ lie in the range $0.71 < \mathcal{H} < 0.84$, with a mean value of $\mathcal{H} = 0.78$ with uncertainty evaluated at the 95% confidence interval of 0.09. The ensemble fractal dimension that corresponds to the total cloud perimeter given by Eq. (11) is $D_e = \mathcal{H} + 1 = 1.78 \pm 0.09$, larger than the canonical value $D \approx 4/3$ often observed for individual clouds obtained using the expression $p \propto \sqrt{a}^D$. Calculated values of $\mathcal{H}$ from the satellite datasets do not appear to depend on the type of satellite orbit or resolution, but they are significantly larger than those found for modeled clouds. $\mathcal{P}_\xi$ measured from SAM follows a power-law with exponent values ranging from $0.60 < \mathcal{H} < 0.71$ depending on threshold value $j$. The average value of $\mathcal{H} = 0.67 \pm 0.08$ is two standard deviations smaller than the the satellite datasets.

Note that modeled values of $\mathcal{H}$ lie closer to the value of 1/3 expected for 3D isotropic turbulence than inferred from the satellite datasets, perhaps reflecting the smaller domain area and atmospheric regime or assumptions used in LES models of subgrid-scale turbulence or local thermodynamic equilibrium. In general, increasing the threshold value $j$ (which determines the minimum vertical cloud thickness required for 2D cloud masking) leads to smaller values of $\mathcal{H}$, reflecting the multifractal nature of clouds. For example, for a detection threshold of $j = 0$, all cloudy pixels in the domain are considered and $\mathcal{H} = 0.71$. Meanwhile, for the highest detection threshold value of $j = 15$, and $\mathcal{H} = 0.61$. The latter case requires that only the largest overlapping cloud structures are included in the analysis, leaving most small, shallow clouds omitted. The smallest clouds are only observed with the finest resolution, resulting in a shallower linear regression slope for more highly thresholded cloud scenes.

## 5 Discussion

To summarize the observations, global cloud perimeter density $\mathcal{P}$ is much more sensitive than cloud fraction $\mathcal{A}$ to measurement resolution $\xi$, but the dependence is also much more simply mathematically characterized. The observed power-law scaling relating $\mathcal{P}$ to $\xi$ is remarkably similar for imagery from a wide range of satellite platforms. We measured an ensemble fractal dimension of $D_e = 1.78 \pm 0.09$, corresponding to a Hurst exponent of $\mathcal{H} = 0.78 \pm 0.09$. Similarly, from DeWitt et al. (2024), $D_e = D\beta \simeq 1.68 \pm 0.06$ derived from satellite observations of the perimeter distribution power-law exponent $\beta = 1.26 \pm 0.06$ and assuming $D = 4/3$ for individual clouds.

To account for how the dimensionality of turbulence may help explain the difference between the measured value of $D_e = 1.78 \pm 0.09$ for satellite observations (Fig. 4) and the theoretical value of 5/3 implied by Eq. (12), we compare $D_e$ with canonical values of $\mathcal{H}$ associated with 2D, 3D, and "intermediate" turbulent regimes and explore "limiting cases" that correspond to possible upper and lower bounds of $\mathcal{P}_\xi$ evaluated at the planetary scale and the Kolmogorov microscale.

## 5.1  Scaling exponents for 3D, 2D, and intermediate turbulence regimes

As introduced in Sect. 1, the theory of 3D isotropic turbulence predicts that the length dependence of turbulent diffusivity follows $K \sim \varepsilon^{1/3} \ell^{4/3}$, i.e., Richardson's 4/3 law. Within the context of resolved clouds, we express $\ell$ as the resolved eddy length $\xi$, assuming that the smallest resolved cloud features are shaped by turbulent eddies of that size. In this case, the turbulent diffusivity scaling expression for cloud edges resolved at scale $\xi$ is

$$K_{\xi,3D} \propto \xi^{4/3} \tag{13}$$

Following from Eq. (3), $K \sim \ell^{1+\mathcal{H}}$, the implied scaling exponent for velocity fluctuations in 3D isotropic turbulence is $\mathcal{H} = 1/3$.

For 2D isotropic turbulence, where vertical motions are negligible, due to e.g., stratification, the diffusivity scaling exponent can be obtained from dimensional analysis with the conserved property being enstrophy $\mathcal{E}$ — the integrated 2D vorticity squared — instead of $\varepsilon$. The dependence of $\mathcal{E}$ on the eddy length scale $\ell$ is $\mathcal{E}(\ell) \sim \Phi^{2/3} \ell^3$ where $\Phi$ is the enstrophy flux density with units s$^{-3}$ (Kraichnan, 1967; Charney, 1971). The velocity scaling exponent is $v \sim \Phi^{1/3} \ell$, and substituting $v$ into $K \sim v\ell$, the 2D turbulent diffusivity scaling becomes

$$K_{\xi,2D} \sim \xi^2 \tag{14}$$

and from Eq. (3), the implied scaling exponent for velocity fluctuations in 2D turbulence is $\mathcal{H} = 1$.

The framework of generalized scale invariance (Schertzer and Lovejoy, 1985) allows for the derivation of an "elliptical dimension" $D_{el}$ that applies to an "intermediate" 23/9D model of anisotropic turbulence at all scales in the atmosphere, rather than distinct regions of 2D isotropic turbulence at large scales and 3D isotropic turbulence at smaller scales. This continuous scaling accounts for the horizontal-vertical anisotropy of the atmosphere due to stratification and is determined by comparing velocity fluctuations $\Delta v_H$ and $\Delta v_V$ corresponding to the horizontal velocity component with subscripts $H$ and $V$ indicating the horizontal or vertical separation between measurements. Horizontal velocity fluctuations have been widely observed to follow a 3D scaling $\Delta v_H \sim \varepsilon^{1/3} \ell^{1/3}$ where $\ell$ ranges from order $\sim 1$ m to the planetary scale (Lovejoy and Schertzer, 2013). The Bolgiano-Obukhov law (Bolgiano Jr, 1959; Obukhov, 1959) describes the corresponding vertical scaling relationship in buoyancy-forced turbulence $\Delta v_V \sim \phi^{1/5} \ell^{3/5}$, where $\phi$ is analogous to the potential energy dissipation rate $\varepsilon$ in the vertical dimension with units m$^2$ s$^{-5}$. To account for this anisotropy in the vertical, $\mathcal{H}_z$ for the combined turbulence case was derived from the ratio of the horizontal and vertical Hurst exponents $\mathcal{H}_H = 1/3$ and $\mathcal{H}_V = 3/5$, resulting in $\mathcal{H}_z = \mathcal{H}_H/\mathcal{H}_V = 5/9 \sim 0.56$. From Eq. (11), the elliptical dimension becomes $D_{el} = 14/9 = 1.56$ (for the volume, $23/9 = 14/9 + 1$. See Lovejoy

(2023) for a review.) From Eq. (3), the turbulent diffusivity for this intermediate 23/9D regime then scales as

$$K_{\xi,int} \sim \xi^{14/9} \tag{15}$$

Note that Eqs. (13) and (14) correspond to the isotropic cases of 2D and 3D turbulence, while Eq. (15) combines the vertical and horizontal components of $\mathcal{H}$ to arrive at an anisotropic case of turbulent diffusivity expression that applies at all scales. Lovejoy et al. (2007) (L07) analyzed 5 m vertical resolution dropsonde wind datasets to determine the relationship $\Delta v_V \sim \ell_V^{\mathcal{H}_V}$ where $\ell_V$ is the vertical separation between measurements. The observed Hurst exponent ranged from $\mathcal{H}_V = 0.60$ — in agreement with the Bolgiano-Obukhov 3/5 scaling for $\ell_V < 1$ km — to $\mathcal{H} = 0.77$ for $\ell_V < 13$ km, the tropospheric depth. Increasing values of $\mathcal{H}_V$ as $\ell_V$ approaches the tropospheric depth were argued to be consistent with more 2D turbulent structures influenced by upper-level jet shear.

Table 3 summarizes previously derived expressions for the scaling exponents $\mathcal{H}$ and $D_e$ for 3D, 2D and intermediate turbulence, along with their relationship to $\mathcal{P}$ through Eq. (8), for comparison with the satellite and numerical model results obtained here. The exponent values in Eqs. (13)-(15) are labeled $D_e$ following Eq. (10).

Observational values from Sect. 4 and from L07 are similar to the theoretically obtained values of $D_e = 5/3$ from Eq. (12) and the 23/9D model implying $D_e = 14/9 = 1.56$, and not to either of 2D or 3D isotropic turbulence. The $D_e = 5/3$ case is closest to the value of $D_e = 1.78 \pm 0.09$ obtained from satellite observations (Fig. 4b) and particularly to the range of values seen in SAM simulations ($1.61 < D_e < 1.72$).

## 5.2 Limiting cases: cloud perimeter density at the turbulent microscale and the planetary scale

Because cloud shapes and sizes are determined by objective physical processes that are independent of subjective measurement resolution, in principle it should be possible to infer information about cloud geometries from the physical properties of the planet and its atmosphere. To this end, we examine order-of-magnitude "limiting case" values for $\mathcal{P}$ evaluated at the smallest and largest possible conceivable scales for clouds, expressed in terms of basic planetary and atmospheric parameters, and compare these with the observations shown in Fig. 4b.

Given the turbulent nature of fractal cloud edges, the Kolmogorov microscale $\eta$ is the smallest theoretical resolution length scale $\xi$, for which $\mathcal{P}$ is anticipated to be a maximum. Substituting $\xi$ with $\eta$ in Eq. (6) yields

$$\mathcal{P}_\eta = \frac{\mathcal{N}\eta}{K_\eta} \tag{16}$$

For the planetary scale (denoted with $\oplus$), Eq. (5) becomes $\mathcal{P}_\oplus \sim \mathcal{N}H/K_\oplus$ where $H$ atmospheric scale height. The planetary-scale diffusivity is $K_\oplus = LU$ where $L = 2a$ and $U = \mathcal{N}H$ is the characteristic speed of the production and dissipation of moist convective potential energy in cloud edge circulations (described in Sect. 2). Thus

$$\mathcal{P}_\oplus \sim \frac{1}{2a} \tag{17}$$

This result is similar to the case where Earth is resolved as a single point source of light, or a "Pale Blue Dot," as coined by Sagan (1994). The extremely idealized case of perimeter density resolved by a single pixel is $\mathcal{P}_\oplus = P_\oplus/A_\oplus$. Considering

**Theory**

| Turbulence Regime | | $\mathcal{H}$ $(\mathcal{P}_\xi \sim \xi^{-\mathcal{H}})$ | $D_e$ $(K_\xi \sim \xi^{-D_e})$ |
|---|---|---|---|
| **3D Isotropic Turbulence** Eq. (13) | | 1/3 (0.33) | 4/3 (1.33) |
| $D_e = 5/3$ Eq. (12) | | 2/3 (0.67) | 5/3 (1.67) |
| **23/9D Elliptical Dimension (GSI)** | | | |
| $\mathcal{H}_H/\mathcal{H}_V$ | Eq. (15) | 5/9 (0.56) | 14/9 (1.56) |
| **2D Turbulence** | Eq. (14) | 1 | 2 |

**Observations**

| Vertical wind structure functions $\mathcal{H}_V$ (L07) | | | |
|---|---|---|---|
| Tropopause: 12.6 km | | 0.77 | 1.77 |
| Surface to 10 km | | 0.60 to 0.75 | |
| Measured cloud perimeters (Figure 4b) | | | |
| Satellite | | $0.78 \pm 0.09$ | 1.78 |
| SAM | | $0.67 \pm 0.08$ | 1.67 |

**Table 3.** Theorized values (top) of $\mathcal{H}$ and $D_e$ from the expressions $\mathcal{P}_\xi \sim \xi^{-\mathcal{H}}$ (Eq. 8), $K_\xi \sim \xi^{D_e}$ (Eq. 10), and $\mathcal{H} = D_e - 1$ (Eq. 11), for the cases of 3D isotropic turbulence, $D_e = 5/3$, the 23/9D elliptical model from generalized scale invariance (GSI), and 2D turbulence. Observations (bottom) include $\mathcal{H}_V$ from Lovejoy et al. (2007) for vertical wind profiles and the measurements obtained here shown in Fig. 4b. Values of $\mathcal{H}$ for each case are compared in Fig. 6. Decimal values are shown alongside the derived fraction values for ease of comparison with observations.

either a square with side length $\xi = 2a$ ($\mathcal{P} = 8a/(4a^2)$) or a circular dot with diameter $2a$ ($\mathcal{P} = 2\pi a/(\pi a^2)$), gives

$$\mathcal{P}_{\oplus,PBD} = \frac{2}{a} \tag{18}$$

In either case, $\mathcal{P}_\oplus$ is a function only of planetary radius $a$. Furthermore, each variable in Eq. (16) can be estimated from basic physical planetary properties, including the atmospheric composition, temperature, and pressure, as described in Appendix A.

## 5.3  Comparison between observations and theory

Figure 5 presents observations and theoretical predictions of $\mathcal{P}_\xi$. Theoretically derived estimates of $\mathcal{P}_\xi$ are obtained from Eq. (8) for three cases: 3D turbulence ($\mathcal{H} = 1/3$ and $D_e = 4/3$), 2D turbulence ($\mathcal{H} = 1$ and $D_e = 2$), and the intermediate case
$D_e = 5/3$. For clarity, the 23/9D case, which has a line nearly the same as the $D_e = 5/3$ case, is not included in Fig. (5). Satellite and SAM measurements are clearly aligned with the case that $D_e = 5/3$, as predicted by Eq. (12), lying distinctly between the curves corresponding to $D_e = 4/3$ for 3D isotropic turbulence and $D_e = 2$ for 2D turbulence. The limiting case for the Kolmogorov microscale $\mathcal{P}_\eta$ marks the intersection of $\mathcal{P}_\xi$ from Eq. (8) where $\xi = \eta$.

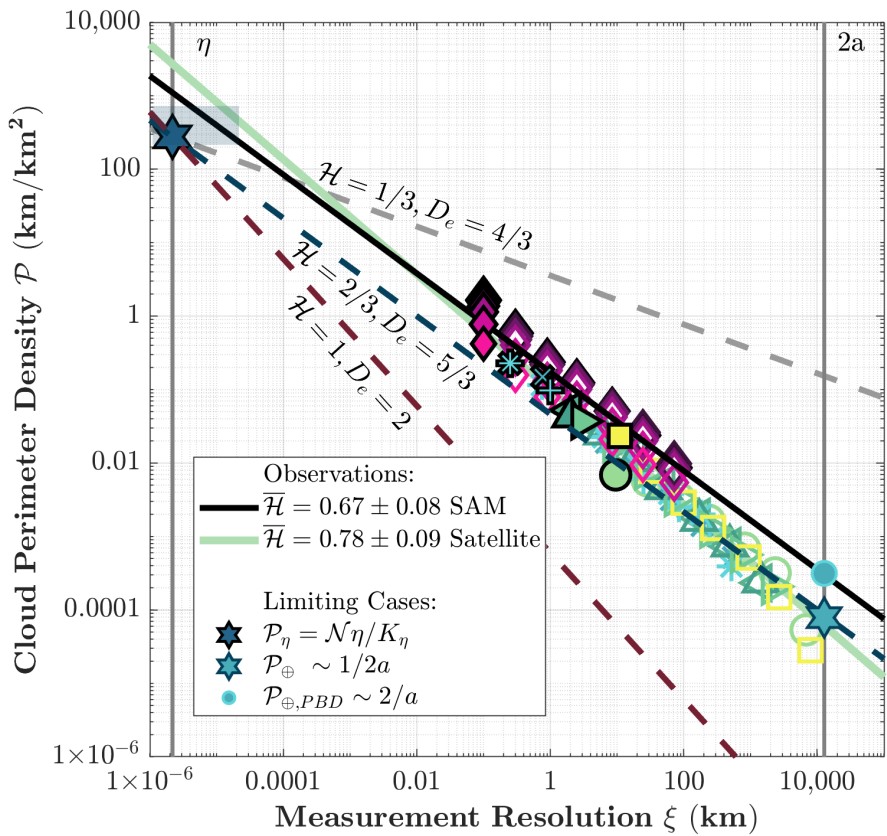

**Figure 5.** Measured perimeter density $\mathcal{P}_\xi$ for the satellites and SAM shown as the same markers from Fig. 4, with the derived $\mathcal{P}_\xi$ from Eq. (8) overlaid as gray ($\mathcal{H} = 1/3$ and $D_e = 4/3$ for 3D turbulence), blue ($D_e = 5/3$), and red ($\mathcal{H} = 1$ and $D_e = 2$ for 2D turbulence) dashed lines. The average scaling exponents $\overline{\mathcal{H}}$ are shown as solid green (satellite) and black (SAM) lines, with the mean and standard deviations in the legend. The limiting case value of $\mathcal{P}_\eta$ from Eq. (16) is shown as a dark blue hexagram with the uncertainty indicated by shading. The limiting case $\mathcal{P}_\oplus$ at the planetary diameter $\xi = 2a$ from Eq. (17) is a light blue hexagram and the Pale Blue Dot case from Eq. (18) is a light blue circle.

What is striking is how well the predicted value of $D_e = 5/3$ connects the highly idealized limiting case values of $\mathcal{P}_\oplus$ and $\mathcal{P}_\eta$ to the observed scaling for $\mathcal{P}_\xi$. The alignment is particularly remarkable considering that $\mathcal{P}_\eta$ and $\mathcal{P}_\oplus$ are obtained only from the physical properties of the planet and its atmosphere, and are separated by 10 orders of magnitude. This correspondence suggests that the statistical aspects of cloud geometries and atmospheric turbulence, $D_e$ and $\mathcal{H}$, could in principle be inferred from knowing only a few basic physical parameters of a planet.

Figure 6 compares the observed and theoretical values of $\mathcal{H}$. The scaling relationship connecting microscale values ($\eta$,$\mathcal{P}_\eta$) to planetary values ($2a$,$\mathcal{P}_\oplus$), as well as the scaling relationships inferred from observations lie between $1/3 < \mathcal{H} < 1$, the limits for 3D and 2D turbulence. The values are most consistent with the case $D_e = 5/3$, and to a lesser extent with the 23/9D intermediate turbulence regime obtained from generalized scale invariance (Schertzer and Lovejoy, 1985; Lovejoy et al., 2007; Lovejoy and Schertzer, 2013) with $\mathcal{H} \sim 5/9$.

Comparing the results here with observations of vertical wind structure functions by Lovejoy et al. (2007) (L07) in Fig. 6 and Table 3, values of $\mathcal{H}$ for the smallest vertical separation distances in L07 ($\ell_V \sim 5$ m, $\mathcal{H} = 0.60$), and for cloud structures resolved vertically in SAM ($\xi_V \sim 100$ m, $\mathcal{H} = 0.67$), correspond most closely to the Bolgiano-Obukov scaling $\mathcal{H} \sim 0.6$. However, the value inferred from satellite observations ($\mathcal{H} = 0.78$) is most consistent with L07 ($\mathcal{H}_V = 0.77$) inferred from vertical separation distances of $\ell_V \sim H$. Despite these variations in $\mathcal{H}$, the observations of clouds reveal an intermediate turbulence regime that excludes both of the purely 2D or 3D isotropic turbulence cases.

## 5.4 Multifractal considerations

Because each of the satellite cloud masks considered in Fig. 4 is generated using a single respective cloud definition threshold, the above analysis is implicitly monofractal. Adopting a monofractal analysis of a field that is multifractal for such quantities as cloud brightness can be problematic. Most importantly, if the cloud brightness field were itself coarsened by averaging over adjacent pixels, the threshold applied to define the presence of a cloud would need to be adjusted to account for the inevitable smoothing of very bright regions with dark regions.

Here, this complication is limited because we are averaging adjacent pixels in a binary cloud mask rather than a brightness field, leading to a more accurate measurement of the fractal dimension (Lovejoy and Schertzer, 1991, Sect. A.4.ii), even as it still does not consider how the fractal dimension varies as a function of threshold. To address this question, we applied various threshold parameters $j$ to define cloudy pixels from the modeled cloud field in SAM. The parameter $j$ is the number of cloudy pixels in each vertical column of the 3D volume required to assign a cloudy pixel in the horizontal 2D cloud mask. The vertical resolution of each pixel is 100 m in the vertical portion of the simulation domain that contains the most cloudy pixels. As shown in Fig. 4a, changing the threshold value of $j$ results in a wide range of horizontal cloud fractions spanning $0.15 < \mathcal{A} < 0.60$. The multifractal nature of clouds is evident in Fig. 6: $\mathcal{H}$ and $D_e$ decrease by 0.11 as the threshold parameter $j$ increases from 1 to 15. What remains clear is that, independent of the threshold considered, the central conclusion of this article remains unchanged, which is that measured values for $\mathcal{H}$ are intermediate to those expected for 2D or 3D isotropic turbulence.

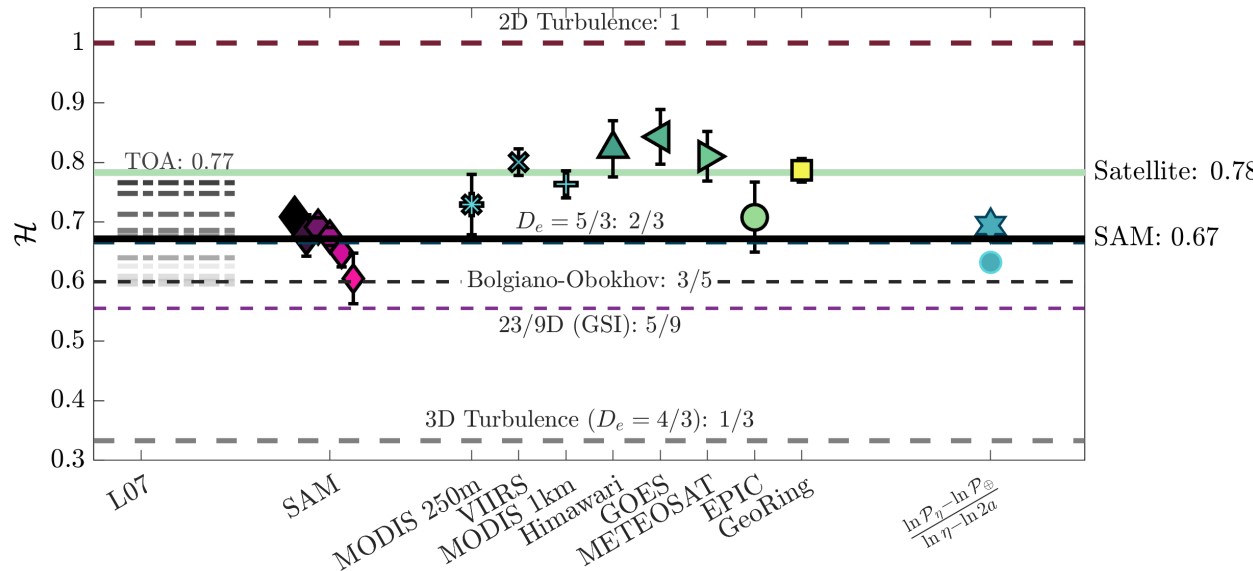

**Figure 6.** Visualization of theorized and observed $\mathcal{H}$. Theorized values of $\mathcal{H}$ are shown as horizontal dashed lines for 2D turbulence (red), for $D_e = 5/3$ (blue), Bolgiano-Obukhov scaling (black), the 23/9D model from generalized scale invariance (GSI) (purple), and 3D turbulence (gray). Observations from Lovejoy et al. (2007) are shown (left) as horizontal gray dashed lines darkening as vertical separation distance $\ell_V$ increases from $\ell_V < 158$ m, to $\ell_V = 12.6$ km corresponding to the top of the atmosphere (TOA). Observations from this work (middle) are shown with symbols corresponding to Figs. 4b and 5 with the averages shown as horizontal green (satellite) and black (SAM) solid lines. On the right are markers corresponding to the slopes from $\mathcal{P}_\eta$ (Eq. 16) to the values of $\mathcal{P}_\oplus$ from Eqs. (17) and (18).

## 6   Conclusions

The measured relationship between the ensemble cloud perimeter density $\mathcal{P}_\xi$ seen from space and the resolution at which it is imaged $\xi$ yields an "ensemble fractal dimension" $D_e$, a scaling exponent analogous to the individual cloud fractal dimension $D$. We conclude that $D_e$ represents the degree to which turbulence is 2D or 3D, and corresponds simply to the Hurst exponent $\mathcal{H}$, the basis of a scaling law for quantifying turbulent fluctuations of atmospheric scalars, through $D_e = \mathcal{H} + 1$.

Global cloud measurements of $\mathcal{P}$ from various satellite orbit types and a Large Eddy Simulation (LES) of tropical convection follow a consistent power-law scaling with respect to $\xi$ across five orders of magnitude. The associated scaling exponent of $\mathcal{H} = 0.78 \pm 0.09$ that we obtained from satellite measurements lies between the theoretical values for isotropic 2D and 3D turbulence, consistent with a model of anisotropic 23/9D turbulence (Schertzer and Lovejoy, 1985).

Measured values of the ensemble fractal dimension $D_e$ are also greater than the value of $D \sim 4/3$ that is often assumed to apply to individual clouds. The value obtained from SAM $D_e = 1.67 \pm 0.08$ is equal to the theorized value of $D_e \sim 5/3$ implied by Eq. (12)). The measured value from satellite imagery $D_e = 1.78 \pm 0.09$ is intermediate to the value of $D_e = 2$ expected for 2D turbulence and $D_e = 4/3$ for 3D turbulence. It is similar to a value of $D_e = 1.68$ suggested by DeWitt et al. (2024) for

cloud ensembles, and to a theoretically derived value of $D_\mu \approx 5/3$ obtained by Hentschel and Procaccia (1984) for intermittent
turbulence. The value of $D_e$ from satellite data is significantly greater than that obtained from analysis of a detailed LES model
of a tropical cloud field, suggesting natural cloud ensembles are more geometrically complex.

Values of $\mathcal{P}$ evaluated at the Kolmogorov microscale $\eta$ and the planetary diameter $2a$ purely from physical parameters lie
remarkably in line with satellite observations and LES model calculations, despite being separated by 10 orders of magnitude in
$\xi$. The value of $\mathcal{P}_\eta$ was only inferred from the molecular composition, temperature, and pressure of clouds and the atmosphere,
while $\mathcal{P}_\oplus$ was inferred from the planetary radius $a$ and the atmospheric depth $H$ and stability $\mathcal{N}$.

Globally distributed, the total perimeter of clouds has a resolution dependence in satellite and numerical datasets, one that
can be tethered to physically parameterized values evaluated at the Kolmogorov microscale and the planetary diameter, that
points to existence of an intermediate 2D/3D turbulence regime that applies at all conceivable tropospheric scales. Observations
of clouds on other planets in the solar system could help identify whether the observed scaling is specific to present-day Earth or
in fact general to stratified atmospheres. Any generalization of the scaling laws could prove useful for constraining predictions
of cloud behaviors in a future climate state on Earth, or for exoplanetary studies where — like Earth's "pale blue dot" — only
coarse-resolution physical parameters are available.

*Code and data availability.* The VIIRS and EPIC datasets were downloaded from NASA Earthdata (https://www.earthdata.nasa.gov/, NASA, 2023) and all others from the ICARE Data Center in Lille, France (https://www.icare. univ-lille.fr/, ICARE, 2023). Code used to analyze data and generate figures is available from the first author upon request.

## Appendix A: Variables and parameters

Values of the parameters and variables used for calculation of $\mathcal{P}$ in Eqs. (8) through (17) are shown in Table A1. Uncertainty in $\mathcal{P}_\eta$ and $\eta$ owes to the range of possible values of $\varepsilon$, $\nu$, and $K_\eta$. Diffusivity and kinematic viscosity are proportional to atmospheric pressure $p$, so uncertainties include the range of values corresponding to the temperature $T$ and $p$ between the surface ($T = 300$ K, $p = 1$ bar) and the top of the troposphere ($T = 200$ K, $p = 0.1$ bar).

The Brunt-Väisälä frequency $\mathcal{N}$ for a dry adiabat is typically expressed as a function of gravity $g$ and vertical temperature profiles, but can also be expressed in terms of physical planetary parameters as $\mathcal{N} \sim g \left( \frac{S(1-\alpha)}{4\sigma} \right)^{-1/8} c_p^{-1/2}$ (Read et al., 2016) where $S$ is the solar constant, $\alpha$ is the planetary albedo, $\sigma$ is the Boltzman constant, and $c_p$ is the specific heat at constant pressure. The value for a moist adiabat shown here is slightly less than, but of the same order of magnitude as the value for a dry adiabat of $0.01$ s$^{-1}$ (Mapes, 2001).

| Parameters | Symbol | Units | Value | Notes |
|---|---|---|---|---|
| Planetary radius | $a$ | km | $6.37 \times 10^3$ | |
| Scale height | $H$ | km | $8.50$ | |
| Brunt-Väisälä frequency | $\mathcal{N}$ | s$^{-1}$ | $6.00 \times 10^{-3}$ | Evaluated for a moist adiabat (Mapes, 2001) |
| Kolmogorov microscale | $\eta$ | km | $2.19 \times 10^{-6}$ | $\eta \sim (\nu^3/\varepsilon)^{1/4}$ |
| TKE dissipation rate | $\varepsilon$ | km$^2$ s$^{-3}$ | $3.00 \times 10^{-9}$ | $10^{-10} < \varepsilon < 10^{-8}$ (Kantha and Hocking, 2011) |
| Kinematic viscosity of air | $\nu$ | km$^2$ s$^{-1}$ | $1.86 \times 10^{-11}$ | $1.5 \times 10^{-11} < \nu < 1.3 \times 10^{-10}$ |
| Diffusion coefficient of air | $K_\eta$ | km$^2$ s$^{-1}$ | $2.42 \times 10^{-11}$ | $2.3 \times 10^{-11} < K_\eta < 9.7 \times 10^{-11}$ (Schwertz and Brow, 1951) |

**Table A1.** Values of variables and parameters described in the text used to determine theoretical values of $\mathcal{P}$ shown in Fig. 5.

*Author contributions.* KNR: methodology, formal analysis and writing (original draft, review and editing). TJG: Conceptualization, funding acquisition, supervision, methodology, writing (review and editing). TDD: methodology and analysis, writing (review and editing). CB: writing (review and editing). SKK: funding acquisition, writing (review and editing). JCR: methodology, writing (review and editing).

*Competing interests.* The authors declare that they have no conflict of interest.

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
