# Peer review of "A global analysis of the fractal properties of clouds revealing anisotropy of turbulence across scales"

_EGUsphere, 2024_

## Referee Comment (RC2)

**Review of Rees et al.,** https://doi.org/10.5194/egusphere-2024-552

GENERAL

This is a thorough study of the scaling characteristics of clouds as observed by satellites, both orbiters and geostationaries. The basic result is important confirmation of the relevance statistcal multifractal studies of atmospheric variables. It is well worthy of publication.

COMMENTARY

Comments are located by line number.

14: The atmosphere is not materially closed. Water has large fluxes in and out, with a residence time of about 10 days. Fluxes of gases such as carbon dioxide, methane, nitrous oxide, halocarbons and ozone also occur, with lifetimes spanning days to centuries. Even the major constituents oxygen ($10^4$ years) and nitrogen ($10^6$ years) are not "materially closed". In addition, volcanology has intermittent effects, and aerosols are injected by various processes, including wave breaking and industrial activity.

15: Inspection of outgoing longwave radiation observed by satellites doesn't look isotropic to this reviewer.

54-56: Restricting characterization to $D$ omits the roles of intermittency $C_1$ and Lévy exponent $\alpha$. That matters especially for water, the material of clouds. See the departures from 5/9 in Figure 13 of https://doi.org/10.3390/meteorology1010003

69: Schertzer and Lovejoy, On the dimension of atmospheric motions, *Turbulence and Chaotic Phenomena in Fluids,* pp.505-508, T. Tatsumi ed., Elsevier North Holland, (1984) deserve at least equal credit with Hentschel and Procaccia and arguably precedence with a 1983 preprint.

Caption, Figure 1. The results of Alder & Wainwright (1970), *Phys. Rev.***1,** 18-21 suggest that isotropic molecular diffusion is never relevant in the atmosphere. See https://doi.org/10.3390/meteorology1010003

165: Large eddy simulation imposes a cubic symmetry on the air that is does not have. What it has is continuous translational symmetry.

248: The polar orbiters are moving at ~7 km/s, unlike the geostationaries.

261: Models assume local thermodynamic equilibrium, which has been argued not to apply. See Figure 3 of *Meteorology* **2023**, *2*(4), 445-463; **https://doi.org/10.3390/meteorology2040026**

284-286: See comment about line 261.

---

## Author Comment (AC1)

The authors thank the reviewers for their very constructive comments. Aside from minor changes and typo corrections, all changes and comment responses are included below. Reviewer comments are shown in **bold** and the author responses are indented. Changes to the manuscript are shown in  and blue text.

**RC1: 'Comment on egusphere-2024-552', Anonymous Referee # 1, 17 May 2024**

**This paper relates theoretically turbulent features and cloud geometry features (mainly perimeter). Then, estimates of the latter on numerous satellite data as well output of numerical simulation enable to confirm previously developed framework to address the issue of anisotropy of turbulence across scales. In general, I found the paper well written and conclusions relevant for the community. I believe that only minor modifications are needed, mainly to improve clarity and help the reader through the numerous equations/approximations and data sets.**

**General comments:**

**- Calling "Xi" (Greek letter) the spatial resolution is a bit confusing, because it often called "scale" with resolution being the ratio between outer scale and observation scales. On a similar point, it is not clear to me why either "l" or "xi" are used whereas it seems to me that they both represent the observation scale at which the studied geometrical set/field is studied. Could you please clarify.**

> The choice of $\xi$ as the variable for spatial resolution follows its use in [1] which relates cloud perimeters to $\xi$ and to the turbulent eddy diffusion. Rather than normalizing by the outer scale $L$, instead $\xi$ is normalized by $\eta$ following e.g., [2, 1] to relate the cloud perimeter to turbulent eddy diffusivity.

> $\xi$ was chosen to link in a straightforward manner cloud perimeters measured by satellites with a range of native spatial resolutions ($\xi_N$) to the fractal dimension through Eqns. (1) and (8), following the "ruler length" definition of $D$ and $D_e$ [4, 3]. $\xi$ is also related to the box-counting dimension where each box is represented by a pixel in a satellite image.

> Revised l. 23-24:

>> (defined as either the pixel side length in a satellite image or the grid spacing in a model following Garrett et al. (2018)).

> Added the following to l. 149-151:

>> Note that $\xi$ is normalized here by $\eta$ rather than its common normalization by an outer scale $L$ (Lovejoy, 2023). We choose this normalization to more conveniently relate $\mathcal{P}_\xi$ to $K_\xi$ and $K_\eta$.

**- A summary with the main formulas that are first theoretically derived and then validated with data would be helpful for the reader. May be in the form of a figure or table.**

> Added the following table summarizing the main formulas:

Table 1: Summary of main formulas

| Equation number | Formula | Reference |
|---|---|---|
| (1) | $p \propto \xi^{1-D}$ | Mandelbrot (1967) |
| (2) | $S(\ell) = \Delta\Theta(\ell) = \langle \Theta(x+\ell) - \Theta(x) \rangle \propto \ell^{\mathcal{H}}$ | Kolmogorov (1941) |
| (3) | $K \sim \ell^{1+\mathcal{H}}$ | Derived from Richardson (1926) and Kolmogorov (1941) |
| (4) | $D = 2 - \mathcal{H}$ | Hentschel and Procaccia (1984) |
| (5) | $K\mathcal{P} = \mathcal{N}\xi_V$ | Garrett et al. (2018) |
| (6) | $\mathcal{P}_\xi \sim \frac{\mathcal{N}\xi}{K_\xi}$ | Equation (5) |
| (7) | $K_\xi = K_\eta \left(\frac{\xi}{\eta}\right)^{1+\mathcal{H}}$ | Krueger et al. (1997) and Eq. (3) |
| (8) | $\mathcal{P}_\xi = \frac{\mathcal{N}\eta}{K_\eta} \left(\frac{\eta}{\xi}\right)^{\mathcal{H}} \propto \xi^{-\mathcal{H}}$ | Equations (6) and (7) |
| (9) | $\mathcal{P}_\xi \propto \xi^{1-D_e}$ | Mandelbrot (1977) |
| (10) | $K_\xi \propto \xi^{D_e}$ | Equations (6) and (9) |
| (11) | $\mathcal{H} = D_e - 1$ | Equations (8) and (9) |
| (12) | $D_e = 3 - D$ | Equations (4) and (11) |

**- It would be interesting to discuss results of Fig. 4 in light of the scaling relation between perimeter and area which is reminded l. 38.**

Revised l. 309-311:

The ensemble fractal dimension that corresponds to the total cloud perimeter given by Eq. (11) is $D_e = \mathcal{H} + 1 = 1.78 \pm 0.09$, larger than the canonical value $D \approx 4/3$ often observed for individual clouds obtained using the expression $p \propto \sqrt{a}^D$.

**- For some of the analysed fields, you have 3D data. Why not trying an analysis in 3D directly instead of reconstructing a 2D field before carrying out the analysis ?**

The next paragraph clarifies that we perform the analysis on generated 2D cloud fields from SAM to represent how the cloud ensembles are viewed by satellite, including any overlap. Added clarification to the lines where SAM is introduced.

Revised l. 231-234:

As a means to compare measurements of $\mathcal{P}_\xi$ from satellite observations to the value derived by Garrett et al. (2018), we  consider the geometries of clouds simulated using the System for Atmospheric Modeling (SAM), calculated as they would be viewed from space. SAM is a high-resolution 3D LES, initialized with idealized GATE Phase III campaign soundings for tropical convection.

**- It should be clarified better how a pixel is set to cloudy or not during the coarsening process. Indeed, many of the observed process depends on this. See detailed comments below.**

Added to Figure 3 (shown below) a diagram to describe the coarsening process with clarifying text added to the caption.

[revised manuscript text omitted]

**- l. 177: the issue of the intermittency correction is briefly mentioned here. There could also be one in eq. 2. I believe that this issue and its implications on the various equations used should be clarified.**

The change made to l. 188-190 in the previous comment also addresses this comment. Additionally, see the revision to l. 68-69:

> Along one dimension $x$, the generalized first-order (which ignores intermittency) "structure function" expresses the covariance of $\Theta$ as a function of separation distance $\ell$.

**- l. 187-190: a scheme on how P and A are computed in practice would be helpful, and also how observation scale is changed.**

Added more detail about how $p$ and $a$ are computed and included a brief example in the revised Figure 3.

Revised l. 203-208:

> The quantities $p$ and $a$ are calculated for all individual  clouds (see Fig. 3f for an example). The perimeter is defined as the sum of all pixel edge lengths along the outer edge of each cloud. Although the example shows that all pixel sizes are equal, in satellite imagery, each pixel has individual values of $\xi_x$ and $\xi_y$ for its width and height, which are adjusted from $\xi_N$ to account for the Earth's curvature away from the satellite nadir vertically and horizontally. The area is the sum of $\xi_x \times \xi_y$ for each pixel in the cloud. For each image, $p$ and $a$ are summed and normalized by domain area $A_d$ to determine $\mathcal{P}$ and $\mathcal{A}$.

**- l. 232-233: what is done once the rounding is implemented ? This approach and its impact should be discussed with regards to a common approach when computing fractal dimension that would be a consider a coarser pixel as cloudy it at least one of the pixels is contains at higher resolution is cloudy.**

This point is clarified in the revised Figure 3.

**- Fig. 3: how are side effect due the round shape handled ?**

The round shape becomes increasingly square during coarsening, and the rounding method simply omits NaN values for pixels beyond the disk. The method of excluding NaNs is sufficient up to coarsening factors $k = \xi/\xi_N \sim 100$. The boxy examples are included to demonstrate the extreme case of coarsening.

Added the clarifying text to the caption of Figure 3:

> Areas outside the circle are marked NaN and so are omitted from the average.

**- In general for section 3: a table with a summary of the data used would be helpful for the reader.**

Added the following table summarizing the satellite datasets used:

Table 2: Summary of satellite datasets used in this study.

| Dataset name | Sensor name | View Type | Approx. nadir resolution | Longitude at nadir | Dates examined | Description of cloud mask algor |
|---|---|---|---|---|---|---|
| **MODIS 250 m** | MODIS | Polar-Orbiting | 250 m | - | 01 January 2021 to 09 January 2021 | DeWitt et al. (2( |
| **VIIRS** | VIIRS | Polar-Orbiting | 750 m | - | 03 June 2021 to 04 June 2021 | Kopp et al. (201 |
| **MODIS 1km** | MODIS | Polar-Orbiting | 1 km | - | 02 June 2012 to 02 June 2012 | Ackerman et al. |
| **Himawari** | AHI | Full-Disk | 2 km | 141° E | 02 June 2021 to 01 July 2021 | Derrien and Glèa |
| **GOES** | ABI | Full-Disk | 2 km | 137° W | 02 June 2021 to 01 July 2021 | Derrien and Glèa |
| **METEOSAT** | SEVIRI | Full-Disk | 3 km | 0° | 02 June 2021 to 01 July 2021 | Derrien and Glèa |
| **EPIC** | EPIC | Full-Disk | 8 km | - | 01 January 2017 to 31 January 2017 | Yang et al. (2019 |
| **GeoRing** | (Composite) | Full-Disk | 11 km | - | 02 June 2021 to 21 June 2021 | Ceamanos et al. |

**- Section 4.1: it is not clear to me why this bifurcation is observed ? How sensitive is it to how a pixel is set to cloudy or not at coarser resolution, which is not very clear for me now ?**

The revision to Figure 3 provides more clarification about how a pixel is set to cloudy or clear during coarsening.

Revised l. 271-275:

This bifurcation of cloud fraction reflects that as an image of a cloud field is coarsened to a single pixel,  the coarsened pixel value is determined by averaging and rounding to zero or unity  the pixel values in the original image (illustrated in Sect. 3). Conversely, a coarsening method in which the presence of any cloudy pixel in the original image results in a cloudy coarsened pixel causes cloud fraction to converge to unity with coarsening (Di Girlamo and Davies, 1997). Due to this averaging method, the value to which cloud fraction bifurcates depends on the native cloud fraction  $\mathcal{A}_N$.

**- Section 4.2 and Fig. 4.b: Can the range of scales used to perform linear be clarified ? Native scale is excluded but seems inserted in straight lines visible in Fig. 4.b. Points for high values of $\xi/\xi_N$ also seem to deviate from the straight line. Indicators of the quality of the linear regression should be added and discussed. Again how sensitive are results to the way a coarser pixel is set to cloudy or not ?**

Added clarification about the linear regression methods and data points used for the fit.

Revised l. 256-259:

A least squares linear regression is performed on values of $\ln \mathcal{P}_\xi$ and $\ln \xi$ to obtain the Hurst exponent $\mathcal{H}$ and $D_e$ from Eqs. (8) and (9). Linear regression was performed on the straightest region of all curves, which was found to be $7 < \xi/\xi_N < 150$ where biases due to interpolation ($\xi/\xi_N < 7$, most significantly for EPIC) and due to the square shape of pixels at very coarse resolutions ($\xi/\xi_N > 150$) are omitted. Uncertainties in the linear regression are evaluated at the 95% confidence level.

Added to the caption of Figure 4:

> Linear regression was performed on the straightest portion of all curves, found to be $7 < \xi/\xi_N < 150$, although the best fit lines are extrapolated to all points to show their relative distance to the fit.

Added to l. 291-293:

> For $\xi > \xi_N$, $\mathcal{P}_\xi$ is well characterized by a linear regression of $\ln \mathcal{P}_\xi$ to $\ln \xi$ (Eq. 8). Linear regression is performed only on data points in the straightest region of all curves, $7 < \xi/\xi_N < 150$, where biases due to interpolation ($\xi/\xi_N < 7$) and the square shape of pixels at very coarse resolutions ($\xi/\xi_N > 150$) are ignored. The lines in the figure are shown extrapolated to all points to demonstrate their relative distance to the least-squares fit, which can be assumed to more reliably reflect the physical fractal nature of the cloud ensemble.

*Citation: https://doi.org/10.5194/egusphere-2024-552-RC1*

**RC2: 'Comment on egusphere-2024-552', Anonymous Referee #2, 27 May 2024**

**GENERAL**

**This is a thorough study of the scaling characteristics of clouds as observed by satellites, both orbiters and geostationaries. The basic result is important confirmation of the relevance statistcal multifractal studies of atmospheric variables. It is well worthy of publication.**

**COMMENTARY**

**Comments are located by line number.**

**14: The atmosphere is not materially closed. Water has large fluxes in and out, with a residence time of about 10 days. Fluxes of gases such as carbon dioxide, methane, nitrous oxide, halocarbons and ozone also occur, with lifetimes spanning days to centuries. Even the major constituents oxygen ($10^4$ years) and nitrogen ($10^6$ years) are not "materially closed". In addition, volcanology has intermittent effects, and aerosols are injected by various processes, including wave breaking and industrial activity.**

Revised l. 15:

The  Earth system is radiatively open and materially closed.

Revised l. 17-18:

Materially, the total dry atmospheric mass is confined to the planet by gravity and can only be redistributed by turbulent circulations that mix air  over a broad range of scales within the thin atmospheric layer.

**15: Inspection of outgoing longwave radiation observed by satellites doesn't look isotropic to this reviewer.**

Revised l. 15-17:

Radiatively, Earth's global mean temperature is sustained by a balance between absorption of high-intensity shortwave sunlight  and the reemission at longwave frequencies to the cold of space.

**54-56: Restricting characterization to D omits the roles of intermittency C1 and Lévy exponent a. That matters especially for water, the material of clouds. See the departures from 5/9 in Figure 13 of https://doi.org/10.3390/meteorology1010003**

Revised l. 55-61:

 The multifractal nature of clouds and their apparent size and type dependence of $D$  seem to contradict the argument that cloud geometries are scale invariant. Additionally, a monofractal  $D$ does not account for multifractal parameters that account for turbulent intermittency (the variability of turbulent fluctuations), notably observed in measurements of water mixing ratio (Tuck, 2022). However, scale invariance might be a reasonable assumption for describing a large ensemble of clouds considered over a sufficiently long period of time and space, especially if turbulent intermittency might be reflected by the geometric intermittency of multiple and varied cloud types in the

ensemble. Indeed, the topic of whether or how scale invariance  applies to atmospheric structures has been the topic of decades of debate (Lovejoy and Schertzer, 2018) .

**69: Schertzer and Lovejoy, On the dimension of atmospheric motions, Turbulence and Chaotic Phenomena in Fluids, pp.505-508, T. Tatsumi ed., Elsevier North Holland, (1984) deserve at least equal credit with Hentschel and Procaccia and arguably precedence with a 1983 preprint.**

Included the reference in l. 73.

**Caption, Figure 1. The results of Alder & Wainwright (1970), Phys. Rev.1, 18-21 suggest that isotropic molecular diffusion is never relevant in the atmosphere. See https://doi.org/10.3390/meteorology1010003**

Added the following to l. 96-98:

Furthermore, the results of Alder and Wainwright (1970) show the formation of vortices even at the $10^{-8}$ m scale, inconsistent with a description of isotropic molecular diffusion (Tuck, 2022).

**165: Large eddy simulation imposes a cubic symmetry on the air that is does not have. What it has is continuous translational symmetry.**

Added the following to l. 179-180:

For comparison with a LES model of a tropical cloud field resolved at 100 m scales, Garrett et al. (2018) applied a value of $\mathcal{H} = 1/3$ to Eq. (7) consistent with Richardson (1926) and the 4/3 law. Implicit in this case is an assumption of 3D isotropic turbulence at resolved scales. The assumption may be appropriate for an LES that chooses a cubic Eulerian grid for computational ease at the expense of losing a Lagrangian perspective.

**248: The polar orbiters are moving at ∼7 km/s, unlike the geostationaries.**

The authors are reluctant to speculate that this is a possible explanation for the differences since EPIC is also nearly geostationary and has results similar to the polar orbiters.

**261: Models assume local thermodynamic equilibrium, which has been argued not to apply. See Figure 3 of Meteorology 2023, 2(4), 445- 463; https://doi.org/10.3390/meteorology2040026**

Added to l. 285-287:

A possible explanation for the discrepancy is that models assume local thermodynamic equilibrium, which has been argued not to apply in the atmosphere (Tuck, 2022).

**284-286: See comment about line 261.**

Revised l. 315-317:

Note that modeled values of $\mathcal{H}$ lie closer to the value of 1/3 expected for 3D isotropic turbulence than  inferred from the satellite datasets, perhaps reflecting the smaller domain area and atmospheric regime or  assumptions used in LES models of subgrid-scale turbulence or local thermodynamic equilibrium.

*Citation: https://doi.org/10.5194/egusphere-2024-552-RC2*

---

## Editor Decision (ED1)

**Comments on A global analysis of the fractal properties of clouds revealing anisotropy of turbulence across scales**

Karlie N. Rees, Timothy J. Garrett, Thomas D. DeWitt, Corey Bois, Steven K. Krueger, and J.r.me C. Riedi

**Overall comments:**

As a preface, I recognize that I am a invested protagonist in the science reported here, so please take these comments as helpful suggestions, not in the spirit of anonymous referee comments.

This paper is a welcome update on a key question of atmospheric dynamics: over what ranges are they scaling? The key finding is that observations of cloud radiances over a huge range of horizontal scales are indeed scaling. This vindicates Richardson's wide range scaling hypothesis updated as confirmed by Lovejoy's 1982 area-perimeter analysis (and numerous spectral and other analyses since). Wide range scaling is incompatible with the still prevalent 2D isotropic/3D isotropic paradigm that necessarily involves a "dimensional transition" somewhere in the mesoscale. The question is which symmetry is dominant: the scale symmetry or the rotational symmetry? Richardson believed it was scaling. Following the isotropic 2D Kraichnan 1968 model, and Charney's 1971 quasi-geostrophic variant, the atmospheric community has largely considered isotropy to be the dominant symmetry, thus implying an elusive dimensional transition/scale break somewhere near the mesoscale. This paper contradicts the latter hypothesis but supports the former. It would be worth bringing this out in the introduction, it will enhance the significance of the work.

My main issue with the paper is that it is monofractal – both in the theoretical model as well as in the data analysis. This aspect with respect to both area-perimeter relations as well as Korcak laws was considered in some detail in several appendices to [*Lovejoy and Schertzer*, 1991]:

http://www.physics.mcgill.ca/~gang/eprints/eprintLovejoy/neweprint/NVAGlovejoyall.pdf

The main conclusion relevant to this paper is that the interpretation of the area-perimeter (A-P) exponent, is quite different when monofractal models (such as fractional Brownian motion), or when multifractal models are used. In the former case (assumed here), the cloud regions exceeding a threshold are assumed to be non fractal (they are assumed to be true (2D) areas, with dimension = 2) so that the usual interpretation of the A-P exponent = D/2 is valid. However if clouds are multifractal, then for any exceedance threshold that define "clouds", the A-P exponent is the ratio D(P)/D(A) of the perimeter dimension D(P) to the fractal dimension of the exceedance set D(A). Since D(P) and D(A) will both decrease with the brightness threshold that defines the sets, the ratio may be quite stable over a range of thresholds, potentially explaining the robustness of the A-P exponent.

The authors may want to reflect and comment on this?

**Minor comments (These are in the attachment).**
(The line numbers are with respect to the second version of the manuscript).

I have few additional minor comments that the authors could address:

Line 55. "The multifractal nature of clouds and their apparent size and type dependence of D seem to contradict the argument that cloud geometries are scale invariant."

This is a nonsequitor: by definition, multifractals are scale invariant. What did you mean to say?

Line 60: "Indeed, the topic of whether or how scale invariance applies to atmospheric structures has been the topic of decades of debate (Lovejoy and Schertzer, 2018)."

In the turbulence community, scale invariance itself is a mainstay for all the theories, the question is the type and range(s) of the scaling: the standard 2D isotropic / 3D isotropic turbulence model with dimensional transition somewhere in the meso-scale versus a single wide range but anisotropic scaling regime (the 23/9D) model. The debate is about the type of scaling: anisotropic or isotropic, the limits of the scaling regime(s) and the values of the scaling exponents.

Note: there is no Lovejoy and Scherzter 2018, you seem to be referring to Lovejoy and Schertzer 2013; please change this throughout the text.

Line 72: Eq. 2 needs an absolute value sign around the difference. In addition, H is only the usual Hurst exponent in the nonintermittent (Gaussian) case. In equation, the H is inspired by Hurst, but is not the same. Also, if fluctuations are defined by other wavelets (i.e. not the differences as indicated), then H can in principle take any real value, the range $-1<H<0$ being particularly important in the macroweather regime.

Line 83: The law eq. 3 ignores intermittency, it is at best an average law. Statistics of other orders will presumably define a hierarchy of (multifractal) exponents. Your mention of the dimensionality is in fact a reference to the 2D isotropic/ 3D isotropic versus 23/9D debate.

Line 94: The correct reference for the spurious nature of the scale breaks in aircraft data is [*Lovejoy et al.*, 2009].

Line 100: The expression "intermediate turbulence regime" is unfortunate since readers will likely think this is a regime intermediate in *spatial scales* whereas I understood (only later in the text) that you meant intermediate in the value of the dimension (i.e. 3>23/9>2). The key point

to make here is that rather than 2 isotropic regimes separated somewhere in the meso-scale, a single (much wider scale range) anisotropic regime was proposed.

Line 104: The 23/9D model proposes that the volume of NONfractal structures scales as $L^{23/9}$. 23/9 is an upper bound on the dimension of the (sparse) fractal structures (i.e. rather than the usual upper bound of D =3). In the 23/9D model, only structures with D<23/9 are fractal.

Also, the exponent is Hz, not H so that it is NOT a Hurst exponent. In the equation "D = 2.55 = 2+H" , H is in fact a RATIO of exponents $H_z$ =$H_{hor}/H_{vertical}$. I'm puzzled because later in the paper, this fact is acknowledged. In terms of the spectral exponents B, the relationship is $H_z$ = $(B_{vertical}$-1)/$(B_{horizontal}$-1) (this is true for both monofractal and multifractal variants of the 23/9D model).

I could also note that the relationship $B = 1+2H$ is only valid for the Gaussian (nonintermittent, nonmultifractal) case (this should be stated), otherwise the are intermittency corrections that are (inconsistently) invoked later (line 111).

Line 110: Eq. 4 applies to the fractal dimension of the geometric set of points on the graph (x,B(x)) where x is the position in a 1-D cloud transect), B is the brightness of 1-D transects through monofractal cloud such as a fractional Brownian motion (fBm) cloud with structure function exponent H. In this case, the fractal dimension of the set of "zero-crossing points" (the intersection of the line B=T = constant with the cloud brightness B(x) is D = 1-H for any threshold T. That is why fBm is a monofractal function. If this fBm model is extended to two dimensional space B(x,y) then the codimension is still H, so that the dimension of the zero-crossing sets (the perimeter set) is independent of the brightness threshold.

Line 150: The nondimensionalization is not only a question of convenience. If the process is multifractal, the key scale is the outer scale and the dissipation plays the role of small cut-off. At any intermediate scale (between the smallest dissipation scale and the outer scale, only the outer scale intervenes, not the inner scale.

Line 184: The intermittency correction arises because turbulence is multifractal, not monofractal.

Line 188: The quantity 3-D is the fractal codimension of a fractal set embedded in a three dimensional space. In (multifractal) turbulence, the codimension is in fact a function (not a unique value) that depends on the threshold used to define the fractal set. At best this equation is useable for a Gaussian model.

Line 354: "Because stratification is only observable in vertical velocity perturbations".
I don't understand: the role of vertical velocity is not clear, and the data on vertical velocities is inadequate. However, the fact of scale dependent stratification and the key $H_z$ parameter ( the ratio of horizontal and vertical scaling exponents) has been estimated in several fields:
Temperature, potential temperature, humidity, horizontal velocity, lidar reflectivity (aerosols), radar reflectivity (clouds). This is reviewed and summarized in ch. 6 of [*Lovejoy and Schertzer*, 2013], see in particular, table 6.5.

Eq. 15: The Richardson law is nearly equivalent to the Kolmogorov law. In the 23/9D model, the standard Kolmogorov law holds in the horizontal (but not vertical), and therefore, we expect the Richardson 4/3 law to hold in the horizontal (but not vertical). Using your eq. 8, we expect the vertical exponent to be 1+3/5 = 8/5 rather than the horizontal value 4/3 (the value 14/9 is not justified). Here you imply the existence of an isotropic Richardson law that would certainly contradict the highly anisotropic 23/9D model.

Line 353: maybe stress that these exponents correspond to the horizontal velocity component with the subscript only indicating the direction of the separation.

References:

Lovejoy, S., and Schertzer, D., Multifractal analysis techniques and the rain and clouds fields from 10$^{-3}$ to 10$^6$m, in *Non-linear variability in geophysics: Scaling and Fractals*, edited by D. Schertzer and S. Lovejoy, pp. 111-144, Kluwer, 1991.
Lovejoy, S., and Schertzer, D., *The Weather and Climate: Emergent Laws and Multifractal Cascades*, 496 pp., Cambridge University Press, 2013.
Lovejoy, S., Tuck, A. F., Schertzer, D., and Hovde, S. J., Reinterpreting aircraft measurements in anisotropic scaling turbulence, *Atmos. Chem. and Phys.* , *9*, 1-19, 2009.

---

## Author Response (AR2)

The authors thank Dr. Lovejoy for the very helpful and constructive comments, which helped improve the accuracy of this article. Aside from minor changes and typo corrections, all changes and comment responses are included below. Editor comments are shown in **bold** and the author responses are indented. Changes to the manuscript are shown in  and blue text. Line numbers correspond to the revised version of the manuscript.

**Editor Report: 'Comment on egusphere-2024-552', Shaun Lovejoy, 26 June 2024**

**Overall comments:**

**As a preface, I recognize that I am a invested protagonist in the science reported here, so please take these comments as helpful suggestions, not in the spirit of anonymous referee comments. This paper is a welcome update on a key question of atmospheric dynamics: over what ranges are they scaling? The key finding is that observations of cloud radiances over a huge range of horizontal scales are indeed scaling. This vindicates Richardson's wide range scaling hypothesis updated as confirmed by Lovejoy's 1982 area-perimeter analysis (and numerous spectral and other analyses since). Wide range scaling is incompatible with the still prevalent 2D isotropic/3D isotropic paradigm that necessarily involves a "dimensional transition" somewhere in the mesoscale. The question is which symmetry is dominant: the scale symmetry or the rotational symmetry? Richardson believed it was scaling. Following the isotropic 2D Kraichnan 1968 model, and Charney's 1971 quasi-geostrophic variant, the atmospheric community has largely considered isotropy to be the dominant symmetry, thus implying an elusive dimensional transition/scale break somewhere near the mesoscale. This paper contradicts the latter hypothesis but supports the former. It would be worth bringing this out in the introduction, it will enhance the significance of the work.**

> Added to ln 132-137
>
> Our findings contradict the theories proposing split 2D and 3D isotropic turbulence regimes separated by a scale break that have prevailed over the past decades (Fiedler and Panofsky, 1970; Nastrom et al., 1984), and support the concept of a wide-ranging, scale invariant 2D-3D  anisotropic turbulence regime proposed by Schertzer and Lovejoy (1985), described in detail by Lovejoy and Schertzer (2013) . We show that this anisotropic turbulence regime applies to cloud perimeters over a remarkable 10 orders of magnitude ranging from the Kolmogorov microscale $\eta$ to the planetary diameter $2a$.

**My main issue with the paper is that it is monofractal – both in the theoretical model as well as in the data analysis. This aspect with respect to both area-perimeter relations as well as Korcak laws was considered in some detail in several appendices to [Lovejoy and Schertzer, 1991]: http://www.physics.mcgill.ca/ gang/eprints/eprintLovejoy/neweprint/NVAGlovejoy-all.pdf The main conclusion relevant to this paper is that the interpretation of the area-perimeter (A-P) exponent, is quite different when monofractal models (such as fractional Brownian motion), or when multifractal models are used. In the former case (assumed here), the cloud regions exceeding a threshold are assumed to be non fractal (they are assumed to be true (2D) areas, with dimension = 2) so that the usual interpretation of the A-P exponent = D/2 is valid. However if clouds are multifractal, then for any exceedance threshold that define "clouds", the A-P exponent is the ratio D(P)/D(A) of the perimeter dimension D(P) to the fractal dimension of the exceedance set D(A). Since D(P) and D(A) will both decrease with the brightness threshold that defines the sets, the ratio may be quite stable over a range of thresholds, potentially explaining the robustness of the A-P exponent. The authors may want to reflect and comment on this?**

The authors thank Dr. Lovejoy for this helpful explanation, and while we do not address this comment specifically in the article because we do not measure the A-P exponent here, we did add clarification on multifractals based on the reference provided.

Note that Section A.4.ii of Lovejoy and Schertzer 1991 suggests that the fractal dimension of perimeters (to include ensembles) can be obtained through coarsening as long as the *set* rather than the *field* is degraded. Rather than degrading the cloud reflectance value from which the cloud mask is determined, we are only degrading the resolution of the binary cloud mask through averaging. As stated, this method should work whether or not the field is multifratcal since it is converted to a set with a well-defined dimension (2 for binary pixels).

We have revised several paragraphs of the introduction to address and clarify these points, while also addressing the first minor comment below:

[revised manuscript text omitted]

**Minor comments (The line numbers are with respect to the second version of the manuscript).:**

**Line 55.** *"The multifractal nature of clouds and their apparent size and type dependence of D seem to contradict the argument that cloud geometries are scale invariant."* **This is a nonsequitor: by definition, multifractals are scale invariant. What did you mean to say?**

See revisions in the previous comment. This statement is removed and clarification is added regarding multifractals and scale invariance.

**Line 60:** *"Indeed, the topic of whether or how scale invariance applies to atmospheric structures has been the topic of decades of debate (Lovejoy and Schertzer, 2018)."* **In the turbulence community, scale invariance itself is a mainstay for all the theories, the question is the type and range(s) of the scaling: the standard 2D isotropic / 3D isotropic turbulence model with dimensional transition somewhere in the meso-scale versus a single wide range but anisotropic scaling regime (the 23/9D) model. The debate is about the type of scaling: anisotropic or isotropic, the limits of the scaling regime(s) and the values of the scaling exponents.**
**Note: there is no Lovejoy and Schertzer 2018, you seem to be referring to Lovejoy and Schertzer 2013; please change this throughout the text.**

Corrected the year in the reference throughout and revised ln 67-73:

> While the fractal dimension and scale invariance are intrinsically linked, their relationship to turbulent structures in the atmosphere is less clear. Two paradigms of turbulence scaling in the atmosphere have been the topic of decades of debate  : split 2D and 3D isotropic scaling regimes for large and small scales (Fiedler and Panofsky, 1970; Nastrom et al., 1984), and wide-ranging anisotropic scaling (Lovejoy, 2023) Both theories originated from the pioneering work of  Richardson (1926), who showed that the turbulent eddy diffusivity $K$, measured using the relative motion of pairs of particles separated by distance $\ell$, followed a power-law with a 4/3 exponent from the millimeter scale for molecular diffusion to the length scale of atmospheric cyclones ($\ell \sim 10^3$ km), $K \propto \ell^{4/3}$, termed the Richardson "4/3 law" of atmospheric diffusion.

**Line 72: Eq. 2 needs an absolute value sign around the difference. In addition, H is only the usual Hurst exponent in the nonintermittent (Gaussian) case. In equation, the H is inspired by Hurst, but is not the same. Also, if fluctuations are defined by other wavelets (i.e. not the differences as indicated), then H can in principle take any real value, the range $-1 < H < 0$ being particularly important in the macroweather regime.**

Added absolute value signs in Eq (2) and specified further about $\mathcal{H}$ in the footnote (ln 80):

> For turbulent scalars, the function tends to be a power-law given by
>
> $$S(\ell) = |\Delta\Theta(\ell)| = \left\langle \left|\Theta(x+\ell) - \Theta(x)\right| \right\rangle \propto \ell^{\mathcal{H}} \tag{2}$$
>
> where brackets indicate averaging over many iterations of the experiment, and $\mathcal{H}$ is the Hurst exponent[1]  with bounds $0 < \mathcal{H} < 1$ (Schertzer and Lovejoy, 1984; Hentschel and Procaccia, 1984; Lovejoy and Schertzer, 2012).
* * *
[1]The Hurst exponent has various mathematical applications, but here we employ its  usage in the field of fractal geometry (for the non-intermittent case) to relate the scaling of turbulent fluctuations with respsect to separation distance $\ell$.

**Line 83: The law eq. 3 ignores intermittency, it is at best an average law. Statistics of other orders will presumably define a hierarchy of (multifractal) exponents. Your mention of the dimensionality is in fact a reference to the 2D isotropic/ 3D isotropic versus 23/9D debate.**

Revised ln 87-92:

The dimensional approximation that $K \sim \ell v$ (Tennekes and Lumley, 1972) results in $K \sim \varepsilon^{1/3}\ell^{4/3}$, reproducing Richardson's 4/3 power-law, and implying that the relationship between diffusivity and the Hurst exponent $\mathcal{H}$ (again ignoring intermittency) follows

$$K \sim \ell^{1+\mathcal{H}} \tag{3}$$

As Sect. 5 elaborates, the value of $\mathcal{H}$  differs based on the dimensionality of the turbulence  (e.g., the case of 2D isotropic turbulence).

**Line 94: The correct reference for the spurious nature of the scale breaks in aircraft data is [Lovejoy et al., 2009]**

Corrected this reference.

**Line 100: The expression *"intermediate turbulence regime"* is unfortunate since readers will likely think this is a regime intermediate in spatial scales whereas I understood (only later in the text) that you meant intermediate in the value of the dimension (i.e. $3 > 23/9 > 2$). The key point to make here is that rather than 2 isotropic regimes separated somewhere in the meso-scale, a single (much wider scale range) anisotropic regime was proposed.**

Revised ln 108-112:

Specifically, Lovejoy et al. (2007) (hereafter L07), and more comprehensively Lovejoy and Schertzer (2013), provided evidence that, rather than two separate isotropic turbulence regimes, the atmosphere is best characterized by a  single anisotropic turbulence regime spanning all scales in the atmosphere. Following the framework of generalized scale invariance (GSI), which accounts for stratification the "23/9D" elliptical model of turbulence in the atmosphere is characterized by a dimension intermediate to 2D and 3D (Schertzer and Lovejoy, 1985).

and ln 129-137:

Section 4 presents the values of the ensemble fractal dimension obtained using several satellite and numerical model datasets. Section 5 interprets the significance of the results by comparing them to the expected values of $D_e$ and $\mathcal{H}$ for 2D and 3D isotropic turbulence, as well as for an  anisotropic turbulence regime that  is intermediate to 2D and 3D at all scales. Our findings contradict the theories of split 2D and 3D isotropic turbulence regimes separated by a scale break, which have prevailed over the past decades (Fiedler and Panofsky, 1970; Nastrom et al., 1984), and support the concept of a wide-ranging, scale invariant 2D-3D  anisotropic turbulence regime proposed by Schertzer and Lovejoy (1985), described in detail by Lovejoy and Schertzer (2013). We show that this anisotropic turbulence regime applies to cloud perimeters over a remarkable 10 orders of magnitude ranging from the Kolmogorov microscale $\eta$ to the planetary diameter $2a$.

**Line 104: The 23/9D model proposes that the volume of NONfractal structures scales as L23/9. 23/9 is an upper bound on the dimension of the (sparse) fractal structures (i.e. rather than the usual upper bound of D =3). In the 23/9D model, only structures with $D < 23/9$ are fractal. Also, the exponent is Hz, not H so that it is NOT a Hurst exponent. In the equation "D = 2.55 = 2+H" , H is in fact a RATIO of exponents Hz =Hhor/Hvertical. I'm puzzled**

**because later in the paper, this fact is acknowledged. In terms of the spectral exponents B, the relationship is Hz = (Bvertical-1)/(Bhorizontal-1) (this is true for both monofractal and multifractal variants of the 23/9D model). I could also note that the relationship B = 1+2H is only valid for the Gaussian (nonintermittent, nonmultifractal) case (this should be stated), otherwise the are intermittency corrections that are (inconsistently) invoked later (line 111).**

> Revised ln 112-118:
>
>> Power spectra of radar reflectivity, cloud radiance, wind speed, and temperature all revealed length-scaling exponents that lie between purely 2D and 3D turbulence cases, consistent with an  anisotropic turbulence regime predicted to have a  volume dimension of  $D = 2.55 = 2 + \mathcal{H}_z$ where $\mathcal{H}_z \approx 0.55$ is the ratio of horizontal and vertical values of $\mathcal{H}$ (discussed further in Sect. 5) (Schertzer and Lovejoy, 1985; Lovejoy and Schertzer, 1985; Lovejoy et al., 1993; Lovejoy, 2021)..  For the Gaussian case, which does not include intermittency or multifractal aspects, $\mathcal{H}$ is calculated from the power spectrum of the observed phenomenon, $E(k) \sim k^{-B}$, where $B = 2\mathcal{H} + 1$. In the 23/9D theory, which incorporates the vertical and horizontal aspects of separation, $\mathcal{H}_z = (B_V - 1)/(B_H - 1)$.

**Line 110: Eq. 4 applies to the fractal dimension of the geometric set of points on the graph (x,B(x)) where x is the position in a 1-D cloud transect), B is the brightness of 1-D transects through monofractal cloud such as a fractional Brownian motion (fBm) cloud with structure function exponent H. In this case, the fractal dimension of the set of "zero-crossing points" (the intersection of the line B=T = constant with the cloud brightness B(x) is D = 1-H for any threshold T. That is why fBm is a monofractal function. If this fBm model is extended to two dimensional space B(x,y) then the codimension is still H, so that the dimension of the zero-crossing sets (the perimeter set) is independent of the brightness threshold.**

> Revised ln 123-125
>
>> Equation (4)  is the 2D analog of the fractal dimension of a geometric set of points. For example, given $(x, \Theta(x))$ where $x$ is the position in a 1D transect and $\Theta$ is the measured cloud brightness, the 1D case $D = 1 - \mathcal{H}$ extends to the 2D cloud perimeter $(\Theta(x,y)$ as $D = 2 - \mathcal{H}$ (Hentschel and Procaccia, 1984).

**Line 150: The nondimensionalization is not only a question of convenience. If the process is multifractal, the key scale is the outer scale and the dissipation plays the role of small cut-off. At any intermediate scale (between the smallest dissipation scale and the outer scale, only the outer scale intervenes, not the inner scale.**

> Removed from ln 23 because we clarify in more detail later
>
>> (defined as either the pixel side length in a satellite image or the grid spacing in a model ).

> Removed the following (ln 165)
>
>>

> and added the following as a footnote:

> Note that $\xi$ is normalized here by $\eta$ rather than the more common normalization by outer scale $L$, the largest eddy of the turbulent flow, from which energy is transferred to smaller eddies of observation scale $\ell = \xi$ in the energy cascade. Because the choice of normalization length scale does not affect calculations of the value of $\mathcal{H}$ or $D_e$, we choose $\eta$ to relate $\mathcal{P}_\xi$ to $K_\xi$ and $K_\eta$. This is consistent with the approach taken by Krueger et al. (1997); Garrett et al. (2018) who focused on the relationship between cloud measurements at scale $\xi$ and turbulent processes at the Kolmogorov microscale $\eta$.

**Line 184: The intermittency correction arises because turbulence is multifractal, not monofractal.**

> Revised ln 195-200:

> > However, while $D = 4/3$ is consistent with values seen for individual clouds, a larger value is required for cloud ensembles, in which case the inequality $D < D_e$ predicted by  Mandelbrot (1977); DeWitt et al. (2024) applies. In a similar adjustment to the individual fractal dimension, Hentschel and Procaccia (1984) related the perimeter fractal dimension of clouds to $\mathcal{H}$ through the expression $D = 2 - \mathcal{H}$ (Eq. 4),  with a correction for  turbulent intermittency ($\mu$, where $D_\mu = (4 + \mu)/3 \approx 5/3$ (described below). We obtain, from Eqs. (11) and (4), an adjustment to $D$ for an ensemble of clouds:
> >
> > $$D_e = 3 - D \qquad (4)$$

**Line 188: The quantity 3-D is the fractal codimension of a fractal set embedded in a three dimensional space. In (multifractal) turbulence, the codimension is in fact a function (not a unique value) that depends on the threshold used to define the fractal set. At best this equation is useable for a Gaussian model.**

> Revised ln 201-202:

> > The quantity $3 - D$ has been  defined as the intermittency exponent by Hentschel and Procaccia (1984) and the multifractal codimension[2] (Schertzer and Lovejoy, 1987) within a 3D  space.

**Line 354: *"Because stratification is only observable in vertical velocity perturbations"*. I don't understand: the role of vertical velocity is not clear, and the data on vertical velocities is inadequate. However, the fact of scale dependent stratification and the key Hz parameter ( the ratio of horizontal and vertical scaling exponents) has been estimated in several fields: Temperature, potential temperature, humidity, horizontal velocity, lidar reflectivity (aerosols), radar reflectivity (clouds). This is reviewed and summarized in ch. 6 of [Lovejoy and Schertzer, 2013], see in particular, table 6.5.**

> Removed this statement. The other points are addressed through minor clarifications in other comments.

**Eq. 15: The Richardson law is nearly equivalent to the Kolmogorov law. In the 23/9D model, the standard Kolmogorov law holds in the horizontal (but not vertical), and therefore, we expect the Richardson 4/3 law to hold in the horizontal (but not vertical). Using your eq. 8, we expect the vertical exponent to be $1+3/5 = 8/5$ rather than the horizontal value 4/3 (the value 14/9 is not justified). Here you imply the existence of an isotropic Richardson law that would certainly contradict the highly anisotropic 23/9D model.**
* * *
[2] The difference between the spatial dimension of the domain and the fractal dimension

Revised ln 374-381:

To account for this anisotropy in the vertical,  $\mathcal{H}_z$ for the combined turbulence case was derived from the ratio of the horizontal and vertical Hurst exponents $\mathcal{H}_H = 1/3$ and $\mathcal{H}_V = 3/5$, resulting in  $\mathcal{H}_z = \mathcal{H}_H/\mathcal{H}_V = 5/9 \sim 0.56$. From Eq. (11), the elliptical dimension becomes $D_{el} = 14/9 = 1.56$ (for the volume, $23/9 = 14/9 + 1$. See Lovejoy (2023) for a review.) From Eq. (3), the turbulent diffusivity for this intermediate 23/9D regime then scales as

$$K_{\xi,int} \sim \xi^{14/9} \tag{5}$$

Note that Eqs. (13) and (14) correspond to the isotropic cases of 2D and 3D turbulence, while Eq. (15) combines the vertical and horizontal components of $\mathcal{H}$ to arrive at an anisotropic case of turbulent diffusivity that applies at all scales.

**Line 353: maybe stress that these exponents correspond to the horizontal velocity component with the subscript only indicating the direction of the separation.**

Revised ln 367-371:

This continuous scaling accounts for the horizontal-vertical anisotropy of the atmosphere due to stratification and is determined by comparing velocity fluctuations $\Delta v_H$ and $\Delta v_V$  corresponding to the horizontal velocity component with subscripts $H$ and $V$ indicating the horizontal or vertical separation between measurements. Horizontal velocity fluctuations have been widely observed to follow a 3D scaling $\Delta v_H \sim \varepsilon^{1/3}\ell^{1/3}$  where $\ell$ ranges from order $\sim$1 m to the planetary scale (Lovejoy and Schertzer, 2013).